# Compress then Merge: From Multiple LoRAs into One Low-Rank Adapter

**Zhengbao He** [1]   **Ruiqi Ding** [1]   **Zhehao Huang** [1]   **Ruikai Yang** [1]   **Tao Li** [1]   **Xiaolin Huang** [1 2]

## Abstract

Low-rank adaptation (LoRA) enables parameter-efficient specialization of foundation models, but the proliferation of task-specific adapters fragments capabilities across many adapters, complicating reuse and deployment. We study the problem of **merging $T$ LoRAs into a single rank-$r$ LoRA**, thereby preserving the benefits of low-rank structure. Existing Merge-then-Compress pipelines treat the rank constraint as an afterthought: they merge adapters in the full parameter space, then compress the merged result to rank $r$ via truncated SVD. However, full-parameter merging may destroy the low-rank structure, making it difficult for subsequent compression to recover an effective rank-$r$ LoRA. We propose Compress-then-Merge (CtM), a reversed pipeline that enforces the rank-$r$ bottleneck *before* merging: CtM computes shared $r$-dimensional subspaces using only the LoRA weights to capture cross-adapter common structure, projects each adapter into the shared subspaces to obtain $r \times r$ coordinates, and then applies standard merging rules in this reduced space. CtM guarantees a rank-$r$ LoRA by construction, avoiding post-hoc truncation, and enables efficient computation in the core space spanned by concatenated LoRA factors. Experiments across multiple models and tasks show that CtM consistently outperforms existing single-LoRA-output baselines while narrowing the performance gap to full-parameter merging methods.

[1]Institute of Image Processing and Pattern Recognition, School of Automation and Intelligent Sensing, Shanghai Jiao Tong University, Shanghai, China [2]Shanghai Key Laboratory of Flexible Medical Robotics, Tongren Hospital, Institute of Medical Robotics, Shanghai Jiao Tong University, Shanghai, China. Correspondence to: Xiaolin Huang <xiaolinhuang@sjtu.edu.cn>.

*Proceedings of the 43$^{rd}$ International Conference on Machine Learning*, Seoul, South Korea. PMLR 306, 2026. Copyright 2026 by the author(s).

## 1. Introduction

Low-Rank Adaptation (LoRA) (Hu et al., 2022) and its variants (Wang et al., 2024b; Li et al., 2025; Zhang et al., 2025) have become a common choice for parameter-efficient fine-tuning (PEFT), enabling practitioners to specialize large foundation models with minimal overhead. This success has created a rapidly growing ecosystem of task-specific LoRA adapters, widely shared on platforms such as HuggingFace (Team, 2024). However, this also fragments capabilities across many adapters, making it difficult to reuse, maintain, and deploy them efficiently. As a result, consolidating knowledge from multiple LoRAs has become an important practical problem and a growing research focus (Panariello et al., 2025; Hu et al., 2025).

Model merging (Yadav et al., 2023; Choi et al., 2024; Yu et al., 2024) provides a natural mechanism for consolidating multiple LoRAs into a single model. Existing approaches (Marczak et al., 2025; Stoica et al., 2025) often merge LoRAs in the full parameter space, producing a full-sized weight update rather than a low-rank adapter. This sacrifices LoRA's key advantages such as modularity and easy reuse. For example, if one later wants to adapt the merged model to a new domain on limited hardware, the merged update needs to remain a single LoRA adapter. In this paper, we focus on a more challenging and practical merging setting, discussed recently by Tang et al., 2025: given $T$ LoRA adapters (trained from the same base model, injected into the same modules, and sharing the same rank), we aim to merge them into **one LoRA adapter** with target rank $r$, i.e., a low-rank update that preserves LoRA's efficiency while integrating capabilities across tasks.

Among merging methods that aim to output a single LoRA, a widely adopted baseline (Tang et al., 2025; Goddard et al., 2024) follows a **Merge-then-Compress** (MtC) pipeline: it first merges multiple LoRAs into a full-sized, high-rank update $\Delta W_{\mathrm{merged}}$, and then compresses it into a rank-$r$ matrix via truncated SVD. This design makes the rank constraint an afterthought: information is unconstrained during merging, but only a small subset survives the later compression. In the view of approximation error measured by Frobenius norm, truncated SVD is indeed optimal. However, the Frobenius norm is not always aligned with average performance across tasks, which can lead to degraded performance and

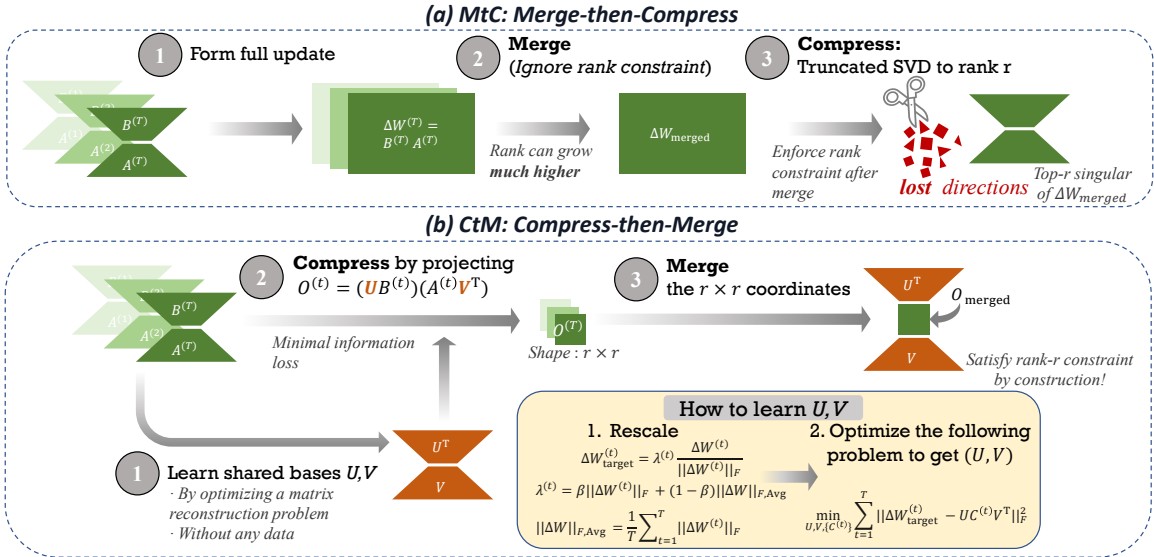

Figure 1. A rank-aware pipeline for merging multiple LoRA adapters into one: Compress-then-Merge (CtM) commits to the rank constraint before merging, avoiding brittle and unstable post-hoc truncation.

hard-to-control multi-task trade-offs. For example, LoRA updates can vary dramatically in scale (Zheng et al., 2025), and SVD-based compression tends to favor large-norm task updates. This can result in performance drops on all tasks but disproportionately larger losses on small-norm ones (as shown in Figure 2). Moreover, LoRA training may introduce *large-magnitude* noisy directions that contribute little to task accuracy (Shuttleworth et al., 2024). Once they enter the top-$r$ subspace, they can consume the rank budget, crowding out other task-relevant components.

More broadly, to produce a single rank-$r$ LoRA from multiple adapters, the final merged results must lie in some $r$-dimensional left/right subspaces. MtC determines the subspaces *after* merging, leaving the subspaces a post-hoc result of the merged result's spectrum. At the same time, the rank-$r$ bottleneck is inherently lossy: once we commit to the final $r$-dimensional subspaces, all components orthogonal to them are irretrievably discarded. This makes the choice of subspace to be the central design consideration, rather than an afterthought determined by the spectrum of a single merged adapter. Recent studies (Kaushik et al., 2025a;b) suggest that models trained on diverse tasks tend to concentrate their weight changes in low-dimensional spectral subspaces, making shared low-dimensional subspaces a plausible and learnable bottleneck for merging. In this paper, we propose a complementary rank-aware pipeline, **Compress-then-Merge** (CtM): we **first** learn shared $r$-dimensional left/right subspaces that capture broadly useful structure across adapters, project each LoRA into the space, and then perform merging directly within it. This design turns subspace selection into an explicit and tunable mechanism, rather than a post-hoc by-product of the merged result.

To obtain the shared subspaces, we optimize a reconstruction objective over the rescaled input LoRA updates, without using auxiliary data. This produces a pair of basis vectors for the left/right subspaces. Crucially, we introduce a rescaling mechanism to avoid the magnitude dominance, ensuring that the learned subspaces reflect broadly useful directions across all tasks. Once the shared space is established, each adapter is projected onto it and represented by a compact $r \times r$ coordinate matrix. We then apply a base merging rule (e.g., TIES, Yadav et al., 2023) directly in this low-dimensional space, and finally lift the merged coordinates back to the original parameter space, which guarantees a rank-$r$ LoRA update by construction. Moreover, by integrating CtM with the core-space framework (Panariello et al., 2025), we obtain a lossless acceleration of the subspace computation. Experiments on vision and language tasks demonstrate consistent improvements over single-LoRA-output approaches.

Our contributions are summarized as follows:

• We identify a key limitation of existing MtC pipelines: the merge process does not take the rank constraint into account, making the subsequent truncated SVD misaligned with the goal of preserving overall task utility, which may lead to unstable and degraded performance.
• We propose CtM, which enforces the rank-$r$ constraint before merging via rescaling-aware shared-subspace learning, and merges adapters in low-dimensional $r \times r$ coordinates, producing a rank-$r$ LoRA update by construction. This makes merging both controllable and efficient, avoiding post-hoc SVD truncation.
• Experiments across multiple architectures and tasks

demonstrate that CtM consistently outperforms existing single-LoRA-output baselines while narrowing the performance gap to full-parameter merging methods. Code is available at https://github.com/ZhengbaoHe/compress-then-merge.

## 2. Related Work

**Model merging.** Model merging consolidates multiple independently trained models into a single model. Most work (Matena & Raffel, 2022; Jin et al., 2023; Nguyen et al., 2025; Jang et al., 2024) focuses on homologous models fine-tuned from the same base checkpoint, where the parameter change is a task vector/matrix $\Delta\theta = \theta_{\text{ft}} - \theta_{\text{base}}$. Task arithmetic (TA, Ilharco et al., 2023) shows that simple averaging can yield a multitask model, and element-wise heuristics such as TIES (Yadav et al., 2023) and DARE (Yu et al., 2024) further reduce interference via trimming, sparsification, and sign-conflict resolution. Recent work studies merging from a matrix/subspace view, e.g., TSV (Gargiulo et al., 2025), Iso-C (Marczak et al., 2025), and CART (Choi et al., 2024), but these methods are primarily designed for full-parameter updates and output dense merged weights.

**Merging methods for LoRA.** A growing body of work studies how to merge LoRA adapters. Alignment-based methods such as KnOTS (Stoica et al., 2025) and CoreSpace (Panariello et al., 2025) build lossless bases to map adapters into a common coordinate system for subsequent merging. However, directly merging can produce a merged update with much higher rank, so they typically rely on an additional SVD-based compression step to generate a fixed-rank merged result (Tang et al., 2025; Goddard et al., 2024). This post-hoc truncation may be misaligned with preserving average multi-task utility under the rank constraint. Several works also aim to produce low-rank adapters directly, including LoRA-LEGO (Zhao et al., 2025), RobustMerge (Zeng et al., 2025), and Iso-CTS (Marczak et al., 2025).

**Beyond training-free consolidation of LoRA.** Beyond training-free merging, some methods learn to select, compose, or distill multiple LoRAs using auxiliary data to consolidate multiple LoRAs (Huang et al., 2023; Wang et al., 2024a; Hu et al., 2025; Alipour & Amiri, 2025; Chen et al., 2025), while others keep multiple LoRA experts and train a router to dynamically select adapters at inference time (Buehler & Buehler, 2024; Feng et al., 2024). These approaches are complementary to ours: they rely on training and/or multi-expert inference, whereas we focus on directly merging into a single fixed-rank LoRA without any training.

**Subspace extraction from multiple LoRAs.** Several works extract a shared low-dimensional structure from collections of LoRA adapters. Compress-then-Serve (Gabrielsson et al., 2025) jointly compresses large numbers of Lo-RAs into shared coordinates/bases to enable efficient multi-adapter serving. EigenLoRAx (Kaushik et al., 2025b) recycles pretrained adapters to identify a principal subspace that can be reused to accelerate adaptation to new tasks with very few additional parameters. These works focus on representing, serving, or reusing *many* adapters, whereas our goal is to merge multiple adapters into a *single* fixed-rank LoRA under a strict rank constraint.

## 3. Merge-then-Compress (MtC) Pipelines

### 3.1. Problem Setup and Notation

**LoRA.** Consider a linear layer with pretrained weight $W_0 \in \mathbb{R}^{n \times m}$. LoRA parameterizes a low-rank update as $\Delta W = BA$, where $B \in \mathbb{R}^{n \times r}$ and $A \in \mathbb{R}^{r \times m}$ with $r \ll \min(m, n)$ (the scaling is omitted since it can be absorbed into $A, B$).

**Merging LoRAs into one LoRA.** We study the merging of $T$ homogeneous LoRA adapters: they are trained from the same base model, injected into the same target layers, and share the same input rank $r_{\text{in}}$. Although we focus on this homogeneous setting for clarity, the formulation can also be extended to heterogeneous ranks or LoRA modules, which will be discussed in the appendix. We perform merging in a layer-wise manner. For a given linear layer, given $T$ LoRA updates $\{(A^{(t)}, B^{(t)})\}_{t=1}^{T}$ with $\Delta W^{(t)} = B^{(t)} A^{(t)}$, our goal is to produce one **low-rank** update $\Delta W_{\text{LoRA}}$ satisfying $\text{rank}(\Delta W_{\text{LoRA}}) \leq r$, without using any additional data. A rank-$r$ LoRA factorization can be obtained from any low-rank decomposition of $\Delta W_{\text{LoRA}}$.

### 3.2. Post-hoc Truncation in MtC

To merge multiple LoRA adapters into a low-rank weight update $\Delta W_{\text{LoRA}}$, a common baseline (which we refer to as **Merge-then-Compress**, MtC) has two steps: **merge** task updates into a full matrix $\Delta W_{\text{merged}}$, then **compress** it into a rank-$r$ matrix via truncated SVD.

The merge step computes each task update $\Delta W^{(t)} = B^{(t)} A^{(t)}$ and applies a merging rule $\mathcal{M}$: $\Delta W_{\text{merged}} = \mathcal{M}\big(\{\Delta W^{(t)}\}_{t=1}^{T}\big)$. Although each $\Delta W^{(t)}$ is low-rank, $\Delta W_{\text{merged}}$ can be much higher-rank (e.g., summation yields $\text{rank}(\Delta W_{\text{merged}}) \leq Tr_{\text{in}}$), often exceeding the target rank $r$. To obtain a rank-$r$ update, prior work (Tang et al., 2025; Goddard et al., 2024) typically applies truncated SVD:

$$\Delta W_{\text{LoRA}} := \Delta \tilde{W}_{\text{merged}} = \sum_{i=1}^{r} \sigma_i u_i v_i^{\top}, \qquad (1)$$

where $(\sigma_i, u_i, v_i)$ are the top-$r$ SVD terms of $\Delta W_{\text{merged}}$.

### 3.3. Limitations of MtC

The main structural issue of MtC is that the rank constraint is enforced only *after* merging. The merge step is free to

accumulate information in all directions, and the subsequent truncated SVD in Eq. 1 keeps only the components that lie in the top-$r$ singular subspace and discards everything orthogonal to it. Although this SVD-based compression is optimal for minimizing Frobenius reconstruction error, this criterion often misaligns with the ultimate goal: average cross-task performance.

Under this post-hoc constraint, any systematic bias in the merged result's spectrum directly determines which directions survive in $\Delta W_{\text{LoRA}}$. A major source of bias is the substantial variation of LoRA update magnitudes across tasks (Zheng et al., 2025): large-norm adapters tend to dominate the spectrum of $\Delta W_{\text{merged}}$, so truncated SVD concentrates rank capacity on these dominant directions, thereby systematically suppresses lower-energy, misaligned components from small-norm adapters. A more subtle failure mode is that "intruder dimensions" may emerge during LoRA training (Shuttleworth et al., 2024): *large-magnitude* directions that appear weakly related to downstream accuracy yet substantially erode pretrained knowledge retention, potentially complicating subsequent merging. Since they also appear as high-energy directions, they are indistinguishable from genuinely useful components and are therefore likely to be preserved by truncated SVD.

As a consequence, MtC can easily over-allocate rank capacity to a few high-magnitude or weakly task-aligned directions, pushing other small-norm, task-relevant updates outside the $r$-dimensional subspace. This can lead to uneven and brittle per-task trade-offs: some tasks are well preserved, while others suffer disproportionate truncation-induced degradation, as illustrated empirically in Figure 2.

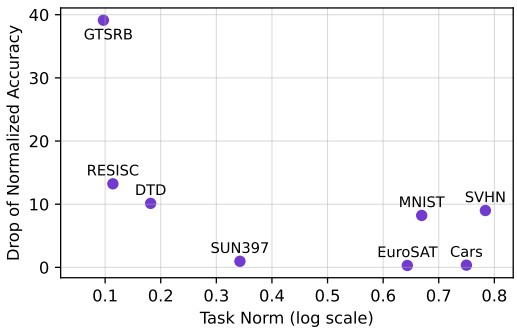

*Figure 2.* Truncation can induce uneven per-task degradation under a fixed rank constraint. We plot each adapter's update norm (log scale) versus the drop of normalized accuracy after truncated SVD. Setup: Iso-C in the full space, following Section 5.

## 4. Compress-then-Merge (CtM) Pipeline

Merging multiple LoRAs into a single rank-$r$ adapter inevitably incurs information loss: any component outside the final $r$-dimensional subspace must be discarded. In

MtC, this loss is imposed only *after* merging and is therefore largely uncontrolled. The truncation subspace is determined by the spectrum of the merged update rather than by cross-task structure. Motivated by this, we move the subspace choice *upfront* and propose the **Compress-then-Merge (CtM)** pipeline. CtM explicitly learns shared $r$-dimensional subspaces to project each LoRA into $r \times r$ coordinates for subsequent merging. The merged result is thus low-rank by construction, mitigating the loss of task-relevant information and removing the need for an additional post-hoc compression step.

### 4.1. Rank-Aware Merging with Shared Subspaces

Our goal is to learn shared $r$-dimensional left/right subspaces with bases $U \in \mathbb{R}^{n \times r}$ and $V \in \mathbb{R}^{m \times r}$ (orthonormal columns) without any auxiliary data, to represent each LoRA in an $r \times r$ coordinate space. We therefore frame the objective as a matrix reconstruction problem: learning low-rank bases to approximate all reweighted LoRAs:

$$\min_{\substack{U^\top U = I_r,\, V^\top V = I_r, \\ \{C^{(t)}\}_{t=1}^T}} \sum_{t=1}^T \left\| \Delta W_{\text{target}}^{(t)} - U C^{(t)} V^\top \right\|_F^2, \quad (2)$$

where $C^{(t)} \in \mathbb{R}^{r \times r}$ is a task-specific dense coordinate matrix in the learned subspaces, $I_r$ is the $r \times r$ identity matrix, and $\Delta W_{\text{target}}^{(t)}$ is a *reweighted surrogate* used *only* for learning $(U, V)$:

$$\Delta W_{\text{target}}^{(t)} = \lambda^{(t)} \frac{\Delta W^{(t)}}{\|\Delta W^{(t)}\|_F}. \quad (3)$$

Normalizing each update $\Delta W^{(t)}$ by its Frobenius norm is a natural way to let all tasks contribute to the subspaces, avoiding the norm-based dominance. However, this also removes any information about how strongly each task tends to modify the base model. To softly reintroduce this scale signal, we attach an importance weight $\lambda^{(t)}$ and define

$$\lambda^{(t)} = \beta \|\Delta W^{(t)}\|_F + (1-\beta) \|\Delta W\|_{F,\text{Avg}},$$
$$\|\Delta W\|_{F,\text{Avg}} = \frac{1}{T} \sum_{t=1}^T \|\Delta W^{(t)}\|_F \quad (4)$$

with $\beta \in [0, 1]$, where $\|\Delta W\|_{F,\text{Avg}}$ denotes the average Frobenius norm over tasks. When $\beta = 0$, all targets share the same norm $\|\Delta W\|_{F,\text{Avg}}$, corresponding to scale-equalized surrogates; while $\beta = 1$ recovers the original updates $\Delta W_{\text{target}}^{(t)} = \Delta W^{(t)}$, but a few large-norm adapters may dominate the learning process. Choosing an intermediate $\beta$ therefore balances equalized contributions with a moderated form of norm-based weighting. Overall, Problem 2 learns $(U, V)$ by jointly balancing the directional structure and a controlled scaling across tasks, aiming to retain information broadly under the rank constraint.

Crucially, the surrogate targets $\Delta W_{\text{target}}^{(t)}$ are used only to learn balanced shared subspaces' bases $(U, V)$. The coordinates for merging are then formed from the *original* LoRAs $\Delta W^{(t)}$ so that merging reflects the true task updates:

$$O^{(t)} = U^\top \Delta W^{(t)} V, \quad t = 1, 2, \ldots, T. \quad (5)$$

We then merge these low-dimensional coordinates using a base merging rule $\mathcal{M}$:

$$O_{\text{merged}} = \mathcal{M}\big(\{O^{(t)}\}_{t=1}^T\big). \quad (6)$$

Finally, we lift the merged coordinate back to the original parameter space:

$$\Delta W_{\text{LoRA}} = U \, O_{\text{merged}} \, V^\top. \quad (7)$$

By construction, $\text{rank}(\Delta W_{\text{LoRA}}) \leq r$, so no additional truncation is required to obtain a rank-$r$ LoRA update.

### 4.2. Solving CtM via Tucker Decomposition

To solve Problem 2, we reformulate it as a Tucker decomposition problem. Specifically, we stack the surrogate targets $\{\Delta W_{\text{target}}^{(t)}\}_{t=1}^T$ along a new mode to form a third-order tensor $\mathcal{X} \in \mathbb{R}^{n \times m \times T}$ whose $t$-th frontal slice is $\mathcal{X}_{[:,:,t]} = \Delta W_{\text{target}}^{(t)}$. Then Problem 2 is equivalent to

$$\min_{\substack{U^\top U = I_r, \, V^\top V = I_r, \\ \mathcal{G} \in \mathbb{R}^{r \times r \times T}}} \left\| \mathcal{X} - \mathcal{G} \times_1 U \times_2 V \times_3 I_T \right\|_F^2, \quad (8)$$

where $\times_i$ denotes the mode-$i$ tensor–matrix product. This is a Tucker-2 decomposition with multilinear rank $(r, r, T)$. Since we do not compress along the task mode, its factor matrix is fixed to the identity $I_T$. Under this formulation, each frontal slice of the core tensor corresponds to a task coefficient matrix, $\mathcal{G}_{[:,:,t]} \equiv C^{(t)}$.

To solve the Tucker decomposition problem, we adopt the standard HOSVD+HOOI strategy: initialize $(U_{(0)}, V_{(0)})$ via Higher-Order Singular Value Decomposition (HOSVD, Lathauwer et al., 2000b), and then refine them by Higher-Order Orthogonal Iteration (HOOI, Lathauwer et al., 2000a). Please refer to Appendix B.2 for details.

### 4.3. Core-Space Acceleration for Subspace Learning

To accelerate the optimization of Problem 2, we adopt the core-space framework (Panariello et al., 2025). This framework compresses the parameter space into a smaller *core space* that exactly preserves all task updates. Concretely, we build two semi-orthogonal bases $U_B \in \mathbb{R}^{n \times (Tr_{\text{in}})}$ and $V_A \in \mathbb{R}^{m \times (Tr_{\text{in}})}$ (i.e., $U_B^\top U_B = V_A^\top V_A = I_{Tr_{\text{in}}}$) that span the union of LoRA left and right factors across tasks. These bases are obtained from the thin SVDs of the concatenated LoRA factors:

$$B_{\text{cat}} = \big[B^{(1)}, \ldots, B^{(T)}\big] = U_B \Sigma_B V_B^\top \in \mathbb{R}^{n \times Tr_{\text{in}}},$$
$$A_{\text{cat}} = \big[A^{(1)}; \ldots; A^{(T)}\big] = U_A \Sigma_A V_A^\top \in \mathbb{R}^{Tr_{\text{in}} \times m}. \quad (9)$$

Using $U_B$ and $V_A$, each update is represented in the *core space* as:

$$\begin{aligned}
M^{(t)} &= \left(U_B^\top B^{(t)}\right) \left(A^{(t)} V_A\right) \\
&= U_B^\top \Delta W^{(t)} V_A \in \mathbb{R}^{(Tr_{\text{in}}) \times (Tr_{\text{in}})},
\end{aligned} \quad (10)$$

and $M_{\text{target}}^{(t)}$ can be computed following Eq. 3-4. For notational simplicity, we denote this coordinate space as $\mathbb{R}^{(Tr_{\text{in}}) \times (Tr_{\text{in}})}$. In practice, $B_{\text{cat}}$ and $A_{\text{cat}}$ are typically full column/row-rank. If rank-deficient, $Tr_{\text{in}}$ can be replaced by the actual rank without affecting the results.

The following theorem formalizes the *lossless* nature of this projection: it preserves every LoRA update exactly, and consequently, the surrogate targets as well (since they differ only by scaling). Proofs are in Appendix A.1.

**Theorem 4.1** (Lossless projection into the core space). *Let $U_B$ and $V_A$ be computed by Eq. 9. Then, for any $t \in \{1, 2, \ldots, T\}$, $\Delta W^{(t)} = U_B U_B^\top \Delta W^{(t)} V_A V_A^\top$ holds.*

Moreover, the next theorem shows that, without loss of optimality, we can restrict our search to solutions with column spaces contained in $\text{Col}(U_B)$ and $\text{Col}(V_A)$. This allows the problem to be reformulated equivalently in the reduced coordinate space later (proof in Appendix A.2).

**Theorem 4.2** (Existence of optimal solutions confined to core space). *There exists an optimal solution $U, V, \{C^{(t)}\}_{t=1}^T$ of Problem 2 satisfying that $\text{Col}(U) \subseteq \text{Col}(U_B)$ and $\text{Col}(V) \subseteq \text{Col}(V_A)$.*

Consequently, we can optimize Problem 2 in the *core space* induced by $(U_B, V_A)$. The original problem is equivalent to the following reduced one:

$$\min_{\substack{U_{\text{core}}^\top U_{\text{core}} = I_r, \\ V_{\text{core}}^\top V_{\text{core}} = I_r, \\ \{C_{\text{core}}^{(t)}\}_{t=1}^T}} \sum_{t=1}^T \left\| M_{\text{target}}^{(t)} - U_{\text{core}} C_{\text{core}}^{(t)} V_{\text{core}}^\top \right\|_F^2, \quad (11)$$

where $U_{\text{core}} \in \mathbb{R}^{(Tr_{\text{in}}) \times r}$, $V_{\text{core}} \in \mathbb{R}^{(Tr_{\text{in}}) \times r}$, and $C_{\text{core}}^{(t)} \in \mathbb{R}^{r \times r}$. After solving Problem 11, the solution is lifted back to the original parameter space via:

$$U = U_B U_{\text{core}}, \qquad V = V_A V_{\text{core}}. \quad (12)$$

Theorem 4.3 below formalizes this equivalence (proof in Appendix A.3). Therefore, we can solve the subspace learning problem in the much smaller core space and then recover a solution in the full parameter space.

**Theorem 4.3** (Equivalence of optimal solutions). *Let $U, V, \{C^{(t)}\}_{t=1}^T$ be an optimal solution of Problem 2 satisfying Theorem 4.2, and define $\tilde{U}_{\text{core}} := U_B^\top U$, $\tilde{V}_{\text{core}} := V_A^\top V$ and $\tilde{C}_{\text{core}}^{(t)} := C^{(t)}$. Then $(\tilde{U}_{\text{core}}, \tilde{V}_{\text{core}}, \{\tilde{C}_{\text{core}}^{(t)}\}_{t=1}^T)$ is an optimal solution of Problem 11. Conversely, for any optimal*

*Table 1.* Normalized accuracies on 8 vision tasks with ViT-B/32. For MtC baselines, we report the average performance without compression, denoted as $\text{Avg}_{\text{full}}$.

| Method | Space | Cars | DTD | EuroSAT | GTSRB | MNIST | RESISC | SUN397 | SVHN | Avg | $\text{Avg}_{\text{full}}$ |
|---|---|---|---|---|---|---|---|---|---|---|---|
| *LoRA's absolute accuracy* | | *74.00* | *58.30* | *99.00* | *92.70* | *99.30* | *88.40* | *64.50* | *96.20* | – | – |
| LoRA-LEGO | – | 81.42 | 72.72 | 47.03 | 38.89 | 50.88 | 69.83 | 97.21 | 40.90 | 62.36 | – |
| RobustMerge | – | 81.38 | 74.91 | 52.67 | 44.71 | 55.15 | 69.56 | 97.17 | 38.94 | 64.31 | – |
| Iso-CTS | – | 81.93 | 76.91 | 42.99 | 53.01 | 59.42 | 74.10 | 98.35 | 43.98 | 66.34 | – |
| | Full | 82.98 | 74.18 | 49.42 | 42.96 | 62.95 | 71.18 | 97.99 | 44.14 | 65.72 | 75.91 |
| | KnOTS | 80.37 | 73.81 | 46.35 | 38.76 | 48.94 | 69.83 | 97.32 | 37.17 | 61.57 | 64.61 |
| Iso-C | CoreSpace | 82.98 | 74.08 | 49.31 | 42.92 | 63.10 | 71.12 | 97.99 | 44.17 | 65.71 | 75.87 |
| | CtM (Ours) | 80.83 | 77.55 | 44.37 | 56.45 | 56.03 | 73.69 | 97.72 | 43.55 | 66.27 | – |
| | Full | 82.51 | 72.90 | 49.94 | 36.93 | 55.41 | 69.22 | 96.95 | 43.15 | 63.38 | 63.94 |
| | KnOTS | 82.35 | 72.99 | 46.61 | 38.66 | 56.50 | 69.27 | 96.44 | 44.22 | 63.38 | 64.40 |
| TIES | CoreSpace | 83.54 | 73.81 | 53.09 | 39.97 | 65.18 | 68.05 | 95.25 | 43.73 | 65.33 | 67.73 |
| | CtM (Ours) | 82.87 | 75.27 | 64.46 | 38.50 | 78.22 | 70.67 | 97.07 | 50.93 | 69.75 | – |
| | Full | 82.51 | 72.81 | 49.57 | 37.03 | 56.12 | 69.31 | 96.90 | 42.75 | 63.37 | 64.03 |
| | KnOTS | 82.35 | 72.99 | 46.61 | 38.66 | 56.50 | 69.27 | 96.44 | 44.22 | 63.38 | 64.55 |
| DARE-TIES | CoreSpace | 84.01 | 73.45 | 53.95 | 40.90 | 64.55 | 69.13 | 96.17 | 45.10 | 65.91 | 69.55 |
| | CtM (Ours) | 83.04 | 75.00 | 64.95 | 38.28 | 79.98 | 69.90 | 96.53 | 57.73 | 70.68 | – |

solution $(U_{\text{core}}, V_{\text{core}}, \{C_{\text{core}}^{(t)}\}_{t=1}^{T})$ *of Problem 11, defining* $\tilde{U} := U_B U_{\text{core}}, \tilde{V} := V_A V_{\text{core}}$ *and* $\tilde{C}^{(t)} := C_{\text{core}}^{(t)}$ *yields an optimal solution of Problem 2.*

Problem 11 can be solved using the same Tucker procedure described in Section 4.2, but with the dimensions reduced from $n \times m$ to $(Tr_{\text{in}}) \times (Tr_{\text{in}})$. Algorithm 1 summarizes the model merging procedure with the core-space acceleration.

**Computational complexity.** We analyze time complexity under thin SVD and assume $(T \cdot r) \ll n$ and $n = m, r_{\text{in}} = r$ for simplicity. Solving Problem 2 in the full space via HOSVD+HOOI costs $\mathcal{O}(n^3 T + n^2 TrK)$, where $K$ denotes the number of HOOI iterations. With the core-space acceleration, constructing $(U_B, V_A)$ and computing $T$ core matrices by Eq. 10 costs $\mathcal{O}(n(Tr)^2)$, and running the same Tucker procedure in the $(Tr) \times (Tr)$ core space costs $\mathcal{O}(T^4 r^3 + T^3 r^3 K)$. Comparing the dominant terms, the core-space formulation thus enjoys approximate speedups:

$$\text{One-off stage: } \frac{n^2}{T \cdot r^2}, \quad \text{Iteration stage: } \frac{n^2}{(Tr)^2}.$$

Detailed derivations are provided in Appendix B.2.

## 5. Experiments

### 5.1. Setup

**Basic setup.** We evaluate merging methods on both *vision* and *language* tasks, following the experimental protocols of KnOTS (Stoica et al., 2025) and CoreSpace (Panariello et al., 2025). For reproducibility, we use the publicly released LoRA checkpoints from KnOTS in the main experiments.

For vision tasks, a CLIP ViT-B/32 model (Radford et al., 2021; Dosovitskiy et al., 2021) is finetuned on eight vision datasets. For language tasks, a LLaMA3-8B (Dubey et al., 2024) is finetuned on six NLI (neural language inference) tasks. All LoRA adapters are applied to attention layers (query, key, value, and output projections) with rank 16.

**Baselines.** We compare our method against both *general-purpose* merging methods under the Merge-then-Compress pipeline and *low-rank-aware* merging methods that directly produce low-rank updates.

For the *general-purpose* methods (**TIES** (Yadav et al., 2023), **DARE** (Yu et al., 2024), and **Iso-C** (Marczak et al., 2025)), we evaluate merging in three coordinate spaces: **Full space**, (the $\Delta W$ space), **KnOTS space** and **CoreSpace** (see Appendix B.1 for detailed description of KnOTS and CoreSpace). After obtaining a merged update in the chosen coordinate space, we map it back to the full space and apply truncated SVD to obtain a low-rank result, following MtC.

The *low-rank-aware* methods (**LoRA-LEGO** (Zhao et al., 2025), **RobustMerge** (Zeng et al., 2025) and **Iso-CTS** (Marczak et al., 2025)) could output a low-rank update directly, thus truncated SVD is not required. For these methods, merging is performed in the full space.

**Evaluation.** Following the widely used validation holdout strategy (Stoica et al., 2025; Panariello et al., 2025; Yadav et al., 2023; Marczak et al., 2025), we tune hyperparameters on a validation holdout and report test performance. We report *normalized accuracy* (each task's test accuracy normalized by its corresponding single-task LoRA). For MtC baselines, we also report the pre-truncation performance,

*Table 2.* Normalized accuracies on 6 NLI tasks with LLaMA3-8B.

| Method | Space | SNLI | MNLI | SICK | QNLI | RTE | SCITAIL | Avg | $\text{Avg}_{\text{full}}$ |
|---|---|---|---|---|---|---|---|---|---|
| *LoRA's absolute accuracy* | | *92.50* | *90.31* | *91.58* | *94.49* | *89.86* | *96.52* | *–* | *–* |
| LoRA-LEGO | – | 87.25 | 92.84 | 87.47 | 58.51 | 99.19 | 95.81 | 86.84 | – |
| RobustMerge | – | 89.14 | 92.13 | 89.56 | 60.98 | 100.00 | 96.20 | 88.00 | – |
| Iso-CTS | – | 90.91 | 87.83 | 85.02 | 68.76 | 100.81 | 95.96 | 88.21 | – |
| Iso-C | Full | 90.91 | 90.06 | 83.51 | 67.60 | 100.81 | 96.64 | 88.25 | 91.38 |
| | KnOTS | 86.75 | 90.07 | 89.12 | 56.48 | 100.00 | 95.57 | 86.33 | 91.99 |
| | CoreSpace | 90.90 | 90.03 | 83.49 | 67.57 | 100.81 | 96.69 | 88.25 | 90.38 |
| | CtM (Ours) | 90.73 | 89.72 | 88.07 | 67.67 | 100.81 | 96.00 | 88.83 | – |
| TIES | Full | 94.78 | 96.16 | 83.42 | 74.04 | 99.19 | 96.69 | 90.71 | 90.73 |
| | KnOTS | 91.66 | 95.05 | 90.34 | 80.43 | 101.61 | 97.32 | 92.74 | 92.90 |
| | CoreSpace | 91.65 | 93.15 | 93.46 | 83.46 | 99.19 | 97.71 | 93.10 | 93.40 |
| | CtM (Ours) | 91.08 | 90.26 | 96.11 | 89.71 | 100.81 | 96.93 | 94.15 | – |
| DARE-TIES | Full | 94.76 | 96.18 | 83.44 | 74.11 | 99.19 | 96.64 | 90.72 | 90.36 |
| | KnOTS | 92.10 | 95.50 | 91.30 | 81.71 | 100.81 | 97.22 | 93.11 | 93.50 |
| | CoreSpace | 91.90 | 91.82 | 95.39 | 80.79 | 100.00 | 97.37 | 92.88 | 93.12 |
| | CtM (Ours) | 92.96 | 94.12 | 97.44 | 94.24 | 97.58 | 97.42 | 95.63 | – |

denoted as $\text{Avg}_{\text{full}}$. Avg and $\text{Avg}_{\text{full}}$ are tuned separately on validation to reflect their respective tuned optimum. Additional details are provided in Appendix C.

## 5.2. Main Results

We report results on two backbones (CLIP ViT-B/32 for multi-task vision and LLaMA3-8B for multi-task NLI) in Table 1 and Table 2. Across both settings, we organize our findings as the following observations.

**Post-hoc compression often incurs substantial performance degradation.** For MtC baselines, the pre-truncation merged update can be strong (reported as $\text{Avg}_{\text{full}}$), but compressing it into rank-$r$ via truncated SVD often leads to a notable degradation (Avg). This truncation gap is most severe in vision tasks. For example, with CoreSpace, performance drops are significant: TIES drops from 67.73 to 65.33, DARE-TIES drops from 69.55 to 65.91, and Iso-C even drops from 75.87 to 65.71. These results demonstrate that a high-quality merged update pre-compression does not guarantee strong performance post-compression. This observation directly motivates CtM, which enforces the rank constraint before merging. For NLI, the truncation gap, while smaller, persists. This indicates that post-compression degradation remains a systematic concern when a fixed rank constraint is required.

**CtM consistently outperforms single-LoRA-output baselines.** Across all general-purpose merging rules evaluated (Iso-C, TIES, and DARE-TIES), CtM consistently achieves better performance than MtC. Results with more base rules are provided in Appendix D.4. In particular, CtM demon-

strates especially strong synergy with TIES-based methods. We hypothesize that this is because, in the compact $r \times r$ coordinate space, task coordinates exhibit lower dispersion, making cross-task conflicts more concentrated and hence easier to resolve. This suggests the potential for developing merging methods better suited to the CtM framework. Overall, on both vision and language tasks, CtM with TIES consistently outperforms all baselines that output a single LoRA, including MtC and low-rank-aware baselines.

**CtM narrows the gap to full-weight merge methods significantly.** On CLIP ViT, CtM with TIES reaches 69.75, which surpasses the best pre-compression performance of TIES and DARE-TIES. Meanwhile, the strongest full-parameter baseline, Iso-C in the full space, achieves $\text{Avg}_{\text{full}} = 75.91$, indicating that full-parameter merging still offers an upper bound without rank constraint, but CtM closes the gap while retaining LoRA's efficiency. On LLaMA NLI, CtM with DARE-TIES attains 95.63, exceeding the corresponding $\text{Avg}_{\text{full}}$ values of all MtC baselines. We speculate that this is because the joint reconstruction objective filters out task-specific noise from individual LoRAs, thereby uncovering an effective shared subspace.

## 5.3. Analysis

**CtM learns a more effective and balanced subspace from both energy and functional perspectives.** To compare the fidelity of the subspaces obtained by different merging pipelines, we extract the left/right subspaces of the merged result $\Delta W_{\text{LoRA}}$ via thin SVD, and project each task LoRA onto these subspaces to obtain its projected LoRA. We then evaluate the projected LoRAs from two complementary

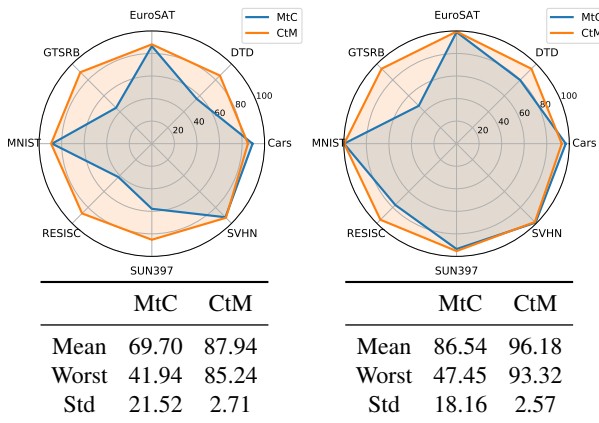

| | MtC | CtM |
|------|-------|-------|
| Mean | 69.70 | 87.94 |
| Worst | 41.94 | 85.24 |
| Std | 21.52 | 2.71 |

| | MtC | CtM |
|------|-------|-------|
| Mean | 86.54 | 96.18 |
| Worst | 47.45 | 93.32 |
| Std | 18.16 | 2.57 |

*Figure 3.* Subspace comparison between CtM and MtC on CLIP ViT. **Left**: energy retention. **Right**: function (accuracy) retention.

perspectives: how much parameter-space energy they retain, and how much task utility they preserve at the functional level (details in Appendix D.1). Figure 3 shows that CtM achieves higher average retention from both perspectives and exhibits substantially lower variance across datasets. This indicates that CtM does not merely improve the mean fidelity of the retained subspace, but also allocates the fixed rank budget more evenly across tasks. In particular, the functional-retention results suggest that CtM is more robust to worst-case task collapse, while MtC's post-hoc truncation may severely discard task-relevant directions for smaller or less dominant adapters. By selecting the shared subspace before merging, CtM mitigates this imbalance and preserves task utility more faithfully under the fixed-rank constraint.

**CtM with SVD-based subspaces.** We further replace the learned shared subspaces in CtM with simple SVD-based ones, obtained by stacking the normalized adapters and taking the top-$r$ left/right singular vectors. As shown in Table 3, this simple variant still outperforms the best MtC baseline, while learned subspaces further improve the performance. These results suggest that CtM does not rely solely on the specific learned-subspace optimization, although better subspace learning provides additional gains.

*Table 3.* CtM with simple SVD-based subspaces still outperforms the best MtC baseline, while learned subspaces further improve performance.

| Method | TIES | DARE-TIES |
|--------|------|-----------|
| Best MtC | 65.33 | 65.91 |
| CtM + SVD subspace | 67.42 | 69.04 |
| CtM + learned subspace | **69.75** | **70.68** |

**Effect of the reweighting hyperparameter $\beta$.** Recall that $\beta$ controls the task-importance weights in Eq. 4, interpolating between scale-equalized targets ($\beta=0$) and scale-preserving targets ($\beta=1$). As shown in Figure 4, CtM is

relatively insensitive to $\beta$ within a reasonable range: a coarse sweep over $\{0, 0.25, 0.5, 0.75\}$ is sufficient in practice. In contrast, setting $\beta=1$ (i.e., no re-normalization) consistently degrades performance, highlighting the necessity of reweighting when learning the shared subspace. We further evaluate $\beta$-reweighting as a plug-in for MtC baselines (Appendix D.3), where it yields limited improvements, and the best reweighted baseline remains substantially below CtM. Overall, these results suggest that CtM's advantage mainly comes from learning shared low-dimensional subspaces *before* merging, rather than from tuning $\beta$.

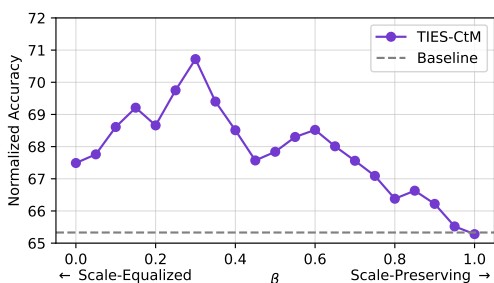

*Figure 4.* Effect of the reweighting hyperparameter $\beta$ of CtM on CLIP ViT. CtM is relatively insensitive to $\beta$ within a reasonable range, so in practice a coarse sweep suffices. Baseline method: TIES in CoreSpace.

**Results of different target ranks.** Figure 5 varies the target rank of the merged adapter while keeping the same eight rank-16 LoRAs as inputs. As expected, the performance of all methods generally improves as the target rank increases, reflecting the benefit of a larger rank budget. CtM achieves the best performance across all ranks, with its advantage over MtC baselines widening at higher ranks. In particular, the MtC variants begin to saturate when rank increases, whereas CtM continues to benefit from additional capacity up to rank 64. These trends suggest that CtM makes more effective use of the available low-rank capacity.

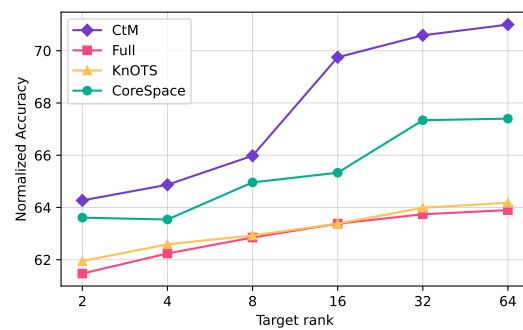

*Figure 5.* Average normalized accuracy under different target ranks on CLIP ViT. The base merging method is TIES.

## 5.4. Additional Experiments

**Core-space acceleration with preserved solution quality.**
Table 4 compares the efficiency of computing the subspaces of CtM in the full parameter space versus in the lossless core space. Optimizing in the core space is substantially faster, as HOSVD/HOOI operate on $(Tr_{\text{in}}) \times (Tr_{\text{in}})$ coordinates rather than $n \times m$ matrices, while final merged performance (Avg. Norm. Acc.) is preserved. Moreover, in two distance metrics applied, the solution obtained in the core space remains close to that from the full-space solution.

*Table 4.* Core-space acceleration for CtM subspace learning on LLaMA3-8B. Besides runtime and average normalized accuracy, we report two distances to the full-space solution: a basis-invariant subspace distance (Chordal) and an element-wise distance.

|                         | Full   | Core     |
| ----------------------- | ------ | -------- |
| Runtime (s)             | 690.83 | 20.77    |
| Avg. Normalized Acc     | 94.15  | 94.15    |
| Chordal Distance to Full | –      | 0.031    |
| Element Distance to Full | –      | 0.000329 |

**Extensions to heterogeneous settings and more LoRAs.**
To test the generalization of CtM, we evaluate it in two *separate* settings: (i) heterogeneous setting, including heterogeneous ranks or injected modules (Appendix D.6) and (ii) a larger collection of LoRAs (Appendix D.7). Results in Table 10/11 and Table 12 indicate that in both scenarios, CtM consistently surpasses the compared single-LoRA-output baselines. This demonstrates its robustness to rank heterogeneity and to an increased number of adapters.

**Results on generative tasks.** We further evaluate CtM beyond the classification-oriented settings considered above. In Appendix D.8, we include additional results on natural language generation with LLaMA2-7B, and in Appendix D.9, we report results on MM-MergeBench, an 8-task multimodal generative benchmark based on LLaVA-1.5-7B. Across these generative settings, CtM consistently improves over the compared baselines, showing that the advantage of CtM extends beyond the classification-oriented settings in the main paper.

## 6. Conclusion

We study the merging of $T$ LoRA adapters into a single LoRA under a rank constraint $r$. We first show that existing MtC pipelines neglect the rank constraint during merging, which can misallocate the limited rank budget and hurt overall multi-task utility. To address this, we propose Compress-then-Merge (CtM), a reversed pipeline that enforces the rank constraint *before* merging. Specifically, CtM first computes shared $r$-dimensional subspaces using only the LoRA weights, projects each adapter into compact $r \times r$ coordinates, and then merges the coordinates directly. Across multi-task vision and language tasks, CtM consistently improves over prior single-LoRA-output baselines and narrows the gap to full-weight methods significantly. Finally, CtM also enables an efficient computation combined with the core-space framework, yielding substantial speedups without sacrificing merged performance. In summary, CtM provides an effective strategy for merging multiple LoRAs into one LoRA under a strict rank constraint.

## Limitations

CtM is designed for the setting of consolidating multiple adapters into a single fixed-rank LoRA, and its effectiveness depends on the existence of shared structure across task updates that can be captured by a common low-rank subspace. When tasks are highly orthogonal, any method constrained to output a single fixed-rank LoRA will inevitably suffer from information loss, and CtM cannot eliminate this fundamental bottleneck. In such cases, a higher rank budget or a more expressive PEFT parameterization for higher rank (Raje et al., 2025; Huang et al., 2025; Singhal et al., 2026) may be required. Moreover, CtM still requires selecting hyperparameters in practice, such as the target rank, the reweighting coefficient, and the base merge rule. Developing more adaptive strategies and merge rules tailored to the CtM coordinate space remains an important direction for future work.

## Acknowledgment

This work was supported by the AI for Science Program, Shanghai Municipal Commission of Economy and Informatization (No. 2025-GZL-RGZN-BTBX-02026) and the National Natural Science Foundation of China (No. 62376155).

## Impact Statement

CtM is intended to improve the efficiency of merging and deploying multiple LoRA adapters, which may be valuable in application scenarios constrained by storage, serving, and management costs. However, in practical use, such methods still involve additional hyperparameter selection and computational overhead, leaving room for improvement in deployment convenience. In addition, merging the capabilities of multiple adapters into a single model can make it more difficult to trace model behavior back to specific data sources or original adapter providers, thereby raising challenges for copyright attribution and model safety auditing. Such methods may also be used to combine multiple specialized adapters with potentially risky capabilities. Therefore, practical deployment should be accompanied by appropriate source management and safety governance measures.

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

# Appendix

# A. Proofs

In this section, we provide the detailed proofs of Theorem 4.1-4.3.

## A.1. Proof of Theorem 4.1.

By construction,

$$B_{\text{cat}} = \left[ B^{(1)}, B^{(2)}, \ldots, B^{(T)} \right] = U_B \Sigma_B V_B^\top,$$

so $\text{Col}(B^{(t)}) \subseteq \text{Col}(B_{\text{cat}}) \subseteq \text{Col}(U_B)$ for every $t$. Since $U_B U_B^\top$ is the orthogonal projector onto $\text{Col}(U_B)$, we have

$$U_B U_B^\top B^{(t)} = B^{(t)} \quad \Rightarrow \quad U_B U_B^\top \Delta W^{(t)} = U_B U_B^\top B^{(t)} A^{(t)} = \Delta W^{(t)}.$$

Similarly,

$$A_{\text{cat}} = \begin{bmatrix} A^{(1)} \\ A^{(2)} \\ \vdots \\ A^{(T)} \end{bmatrix} = U_A \Sigma_A V_A^\top$$

implies $\text{Row}(A^{(t)}) \subseteq \text{Row}(A_{\text{cat}}) \subseteq \text{Col}(V_A)$, hence $A^{(t)} V_A V_A^\top = A^{(t)}$ and

$$\Delta W^{(t)} V_A V_A^\top = B^{(t)} A^{(t)} V_A V_A^\top = B^{(t)} A^{(t)} = \Delta W^{(t)}.$$

$\square$

## A.2. Proof of Theorem 4.2.

***Proof sketch.*** We first show that, for any optimal solution of Problem 2, the reconstructed matrices $F^{(t)} = U C^{(t)} V^\top$ all lie in the subspace $\{ U_B Y V_A^\top : Y \in \mathbb{R}^{(T r_{\text{in}}) \times (T r_{\text{in}})} \}$ induced by $U_B$ and $V_A$. Then we re-factorize the optimal $\{ F^{(t)} \}_{t=1}^T$ within this subspace to obtain a new optimal solution $(U, V, \{C^{(t)}\})$ such that $\text{Col}(U) \subseteq \text{Col}(U_B)$ and $\text{Col}(V) \subseteq \text{Col}(V_A)$, which proves the lemma.

***Detailed proof.*** Firstly, we define the following linear map using $U_B$ and $V_A$:

$$\mathcal{P} : \mathbb{R}^{n \times m} \to \mathbb{R}^{n \times m}, \quad \mathcal{P}(X) := U_B U_B^\top X V_A V_A^\top.$$

According to Theorem 4.1, for $t \in \{1, \ldots, T\}$, the target matrices $\Delta W_{\text{target}}^{(t)}$ lie in the subspace $\text{Im}(\mathcal{P})$, i.e.,

$$\Delta W_{\text{target}}^{(t)} = \mathcal{P}\left( \Delta W_{\text{target}}^{(t)} \right) = U_B U_B^\top \Delta W_{\text{target}}^{(t)} V_A V_A^\top.$$

Let $(U^\star, V^\star, \{C^{(t)\star}\}_{t=1}^T)$ be an arbitrary optimal solution of Problem 2, and define

$$F^{(t)\star} := U^\star C^{(t)\star} V^{\star\top} \in \mathbb{R}^{n \times m}, \qquad t = 1, \ldots, T.$$

Then we decompose $F^{(t)\star}$ into its components parallel and orthogonal to $\text{Im}(\mathcal{P})$:

$$F^{(t)\star} = \mathcal{P}\left( F^{(t)\star} \right) + (I - \mathcal{P})\left( F^{(t)\star} \right) =: F_{\parallel}^{(t)} + F_{\perp}^{(t)},$$

where $F_{\parallel}^{(t)} \in \text{Im}(\mathcal{P})$ and $F_{\perp}^{(t)} \in \text{Im}(\mathcal{P})^\perp$.

Since $\Delta W_{\text{target}}^{(t)} \in \text{Im}(\mathcal{P})$ and $\mathcal{P}$ is an orthogonal projector with respect to the Frobenius inner product, we have

$$\Delta W_{\text{target}}^{(t)} - F_{\parallel}^{(t)} \in \text{Im}(\mathcal{P}), \qquad F_{\perp}^{(t)} \in \text{Im}(\mathcal{P})^\perp,$$

and hence

$$\left\langle \Delta W_{\text{target}}^{(t)} - F_{\parallel}^{(t)}, F_{\perp}^{(t)} \right\rangle_F = 0.$$

Therefore,

$$\left\| \Delta W_{\text{target}}^{(t)} - F^{(t)\star} \right\|_F^2 = \left\| \Delta W_{\text{target}}^{(t)} - F_{\parallel}^{(t)} - F_{\perp}^{(t)} \right\|_F^2$$
$$= \left\| \Delta W_{\text{target}}^{(t)} - F_{\parallel}^{(t)} \right\|_F^2 + \left\| F_{\perp}^{(t)} \right\|_F^2.$$

Summing over $t = 1, \ldots, T$ yields

$$\sum_{t=1}^{T} \left\| \Delta W_{\text{target}}^{(t)} - F^{(t)\star} \right\|_F^2 = \sum_{t=1}^{T} \left\| \Delta W_{\text{target}}^{(t)} - F_{\parallel}^{(t)} \right\|_F^2 + \sum_{t=1}^{T} \left\| F_{\perp}^{(t)} \right\|_F^2.$$

By definition,

$$\sum_{t=1}^{T} \left\| \Delta W_{\text{target}}^{(t)} - F^{(t)\star} \right\|_F^2$$

is the optimal objective value of Problem 2. The above expression shows that replacing $\{F^{(t)\star}\}$ by their projections $\{F_{\parallel}^{(t)}\}$ cannot increase the objective value. Hence, optimality of $(U^\star, V^\star, \{C^{(t)\star}\})$ implies

$$\sum_{t=1}^{T} \left\| F_{\perp}^{(t)} \right\|_F^2 = 0,$$

and therefore $F_{\perp}^{(t)} = 0$ for all $t$. In particular,

$$F^{(t)\star} = \mathcal{P}(F^{(t)\star}) = U_B U_B^\top F^{(t)\star} V_A V_A^\top \in \text{Im}(\mathcal{P}) \quad \text{for all } t.$$

Thus, for each $t$ there exists a matrix $Y^{(t)} \in \mathbb{R}^{(Tr_{\text{in}}) \times (Tr_{\text{in}})}$ such that

$$F^{(t)\star} = U_B Y^{(t)} V_A^\top.$$

Let

$$\mathcal{U} := \text{span}\{\text{Col}(F^{(t)\star}) : t = 1, \ldots, T\} \subseteq \text{Col}(U_B),$$
$$\mathcal{V} := \text{span}\{\text{Row}(F^{(t)\star}) : t = 1, \ldots, T\} \subseteq \text{Col}(V_A).$$

Since each $F^{(t)\star} = U^\star C^{(t)\star} V^{\star\top}$ has rank at most $r$ and shares the common left and right factors $U^\star$ and $V^\star$, we have $\dim(\mathcal{U}) \le r$ and $\dim(\mathcal{V}) \le r$.

Choose $r$-dimensional subspaces $\tilde{\mathcal{U}}, \tilde{\mathcal{V}}$ such that $\mathcal{U} \subseteq \tilde{\mathcal{U}} \subseteq \text{Col}(U_B)$ and $\mathcal{V} \subseteq \tilde{\mathcal{V}} \subseteq \text{Col}(V_A)$. Let $U \in \mathbb{R}^{n \times r}$ and $V \in \mathbb{R}^{m \times r}$ are semi-orthogonal matrices whose columns form bases of $\tilde{\mathcal{U}}$ and $\tilde{\mathcal{V}}$, respectively. Then, for each $t$, the matrix $F^{(t)\star}$ has its column space contained in $\tilde{\mathcal{U}}$ and its row space contained in $\tilde{\mathcal{V}}$, so there exists $C^{(t)} \in \mathbb{R}^{r \times r}$ such that

$$F^{(t)\star} = U C^{(t)} V^\top.$$

In fact, these coefficients can be chosen explicitly as

$$C^{(t)} = U^\top F^{(t)\star} V, \qquad t = 1, \ldots, T.$$

Now we have a feasible solution for Problem 2:

$$U, V, \{C^{(t)}\}_{t=1}^{T}.$$

For each $t$, we still have $U C^{(t)} V^\top = F^{(t)\star}$. Hence the objective value achieved by $(U, V, \{C^{(t)}\})$ is exactly the same as that of $(U^\star, V^\star, \{C^{(t)\star}\})$, i.e., it is optimal. Moreover, $\text{Col}(U) = \tilde{\mathcal{U}} \subseteq \text{Col}(U_B)$ and $\text{Col}(V) = \tilde{\mathcal{V}} \subseteq \text{Col}(V_A)$ by construction.

$\square$

## A.3. Proof of Theorem 4.3.

**Lemma A.1** (Loss-preserving reparameterization). *Let $\mathcal{F}_{\text{res}}$ be the feasible set of Problem 2 restricted by Theorem 4.2, and let $\mathcal{F}_{\text{core}}$ be the feasible set of Problem 11. Define the mappings*

$$\Phi : \mathcal{F}_{\text{res}} \to \mathcal{F}_{\text{core}}, \quad (U, V, \{C^{(t)}\}) \mapsto (\tilde{U}_{\text{core}}, \tilde{V}_{\text{core}}, \{\tilde{C}_{\text{core}}^{(t)}\}) := (U_B^\top U, V_A^\top V, \{C^{(t)}\}),$$

$$\Psi : \mathcal{F}_{\text{core}} \to \mathcal{F}_{\text{res}}, \quad (U_{\text{core}}, V_{\text{core}}, \{C_{\text{core}}^{(t)}\}) \mapsto (\tilde{U}, \tilde{V}, \{\tilde{C}^{(t)}\}) := (U_B U_{\text{core}}, V_A V_{\text{core}}, \{C_{\text{core}}^{(t)}\}).$$

*Then $\Phi$ and $\Psi$ are well-defined inverses of each other, and for all feasible tuples,*

$$\mathcal{L}(U, V, \{C^{(t)}\}) = \mathcal{L}_{\text{core}}\big(\Phi(U, V, \{C^{(t)}\})\big), \qquad \mathcal{L}_{\text{core}}(U_{\text{core}}, V_{\text{core}}, \{C_{\text{core}}^{(t)}\}) = \mathcal{L}\big(\Psi(U_{\text{core}}, V_{\text{core}}, \{C_{\text{core}}^{(t)}\})\big).$$

**Proof.** According to the definition of $M_{\text{target}}^{(t)}$ and the proof of Theorem 4.2, we have for each $t$ that

$$\Delta W_{\text{target}}^{(t)} = U_B M_{\text{target}}^{(t)} V_A^\top.$$

We first recall the shapes and orthogonality of the matrices $U, V, U_B, V_A, U_{\text{core}}, V_{\text{core}}$:

$$U \in \mathbb{R}^{n \times r}, \quad U^\top U = I_r,$$
$$V \in \mathbb{R}^{m \times r}, \quad V^\top V = I_r,$$
$$U_B \in \mathbb{R}^{n \times (Tr_{\text{in}})}, \quad U_B^\top U_B = I_{Tr_{\text{in}}},$$
$$V_A \in \mathbb{R}^{m \times (Tr_{\text{in}})}, \quad V_A^\top V_A = I_{Tr_{\text{in}}},$$
$$U_{\text{core}} \in \mathbb{R}^{(Tr_{\text{in}}) \times r}, \quad U_{\text{core}}^\top U_{\text{core}} = I_r,$$
$$V_{\text{core}} \in \mathbb{R}^{(Tr_{\text{in}}) \times r}, \quad V_{\text{core}}^\top V_{\text{core}} = I_r.$$

**From the original problem to the core problem.** Let $(U, V, \{C^{(t)}\})$ be any feasible solution of Problem 2 satisfying $\text{Col}(U) \subseteq \text{Col}(U_B)$ and $\text{Col}(V) \subseteq \text{Col}(V_A)$. There exist matrices $R_U, R_V \in \mathbb{R}^{(Tr_{\text{in}}) \times r}$ such that

$$U = U_B R_U, \qquad V = V_A R_V.$$

Using the orthogonality constraints, we obtain

$$I_r = U^\top U = R_U^\top U_B^\top U_B R_U = R_U^\top R_U, \qquad I_r = V^\top V = R_V^\top V_A^\top V_A R_V = R_V^\top R_V.$$

Hence $R_U$ and $R_V$ are semi-orthogonal matrices. By definition of $\tilde{U}_{\text{core}}$ and $\tilde{V}_{\text{core}}$,

$$\tilde{U}_{\text{core}} = U_B^\top U = U_B^\top U_B R_U = R_U, \qquad \tilde{V}_{\text{core}} = V_A^\top V = V_A^\top V_A R_V = R_V,$$

and therefore $\tilde{U}_{\text{core}}^\top \tilde{U}_{\text{core}} = I_r$ and $\tilde{V}_{\text{core}}^\top \tilde{V}_{\text{core}} = I_r$. Thus $(\tilde{U}_{\text{core}}, \tilde{V}_{\text{core}}, \{\tilde{C}_{\text{core}}^{(t)}\})$ with $\tilde{C}_{\text{core}}^{(t)} := C^{(t)}$ is feasible for Problem 11.

Next we show that the per-task losses coincide. For any $t$,

$$\begin{aligned}
\mathcal{L}^{(t)}(U, V, C^{(t)}) &= \big\|\Delta W_{\text{target}}^{(t)} - U C^{(t)} V^\top\big\|_F^2 \\
&= \big\|U_B M_{\text{target}}^{(t)} V_A^\top - U_B \tilde{U}_{\text{core}} \tilde{C}_{\text{core}}^{(t)} \tilde{V}_{\text{core}}^\top V_A^\top\big\|_F^2 \\
&= \big\|U_B \big(M_{\text{target}}^{(t)} - \tilde{U}_{\text{core}} \tilde{C}_{\text{core}}^{(t)} \tilde{V}_{\text{core}}^\top\big) V_A^\top\big\|_F^2.
\end{aligned}$$

Since $U_B$ and $V_A$ have orthonormal columns, multiplication by $U_B$ on the left and $V_A^\top$ on the right preserves the Frobenius norm, therefore,

$$\mathcal{L}^{(t)}(U, V, C^{(t)}) = \big\|M_{\text{target}}^{(t)} - \tilde{U}_{\text{core}} \tilde{C}_{\text{core}}^{(t)} \tilde{V}_{\text{core}}^\top\big\|_F^2 = \mathcal{L}_{\text{core}}^{(t)}(\tilde{U}_{\text{core}}, \tilde{V}_{\text{core}}, \tilde{C}_{\text{core}}^{(t)}).$$

Summing over $t$ yields

$$\mathcal{L}(U, V, \{C^{(t)}\}) = \mathcal{L}_{\text{core}}(\tilde{U}_{\text{core}}, \tilde{V}_{\text{core}}, \{\tilde{C}_{\text{core}}^{(t)}\}).$$

**From the core problem to the original problem.** Conversely, let $(U_{\text{core}}, V_{\text{core}}, \{C_{\text{core}}^{(t)}\})$ be any feasible solution of Problem 11, and define

$$\tilde{U} := U_B U_{\text{core}}, \qquad \tilde{V} := V_A V_{\text{core}}, \qquad \tilde{C}^{(t)} := C_{\text{core}}^{(t)}.$$

We have:

$$\tilde{U}^\top \tilde{U} = (U_B U_{\text{core}})^\top U_B U_{\text{core}} = U_{\text{core}}^\top U_B^\top U_B U_{\text{core}} = I_r,$$
$$\tilde{V}^\top \tilde{V} = (V_A V_{\text{core}})^\top V_A V_{\text{core}} = V_{\text{core}}^\top V_A^\top V_A V_{\text{core}} = I_r.$$

So $(\tilde{U}, \tilde{V}, \{\tilde{C}^{(t)}\})$ is feasible for Problem 2. Moreover,

$$\begin{aligned}
\mathcal{L}^{(t)}(\tilde{U}, \tilde{V}, \tilde{C}^{(t)}) &= \big\| \Delta W_{\text{target}}^{(t)} - \tilde{U} \tilde{C}^{(t)} \tilde{V}^\top \big\|_F^2 \\
&= \big\| U_B M_{\text{target}}^{(t)} V_A^\top - U_B U_{\text{core}} C_{\text{core}}^{(t)} V_{\text{core}}^\top V_A^\top \big\|_F^2 \\
&= \big\| M_{\text{target}}^{(t)} - U_{\text{core}} C_{\text{core}}^{(t)} V_{\text{core}}^\top \big\|_F^2 \\
&= \mathcal{L}_{\text{core}}^{(t)}(U_{\text{core}}, V_{\text{core}}, C_{\text{core}}^{(t)}),
\end{aligned}$$

and summing over $t$ yields

$$\mathcal{L}(\tilde{U}, \tilde{V}, \{\tilde{C}^{(t)}\}) = \mathcal{L}_{\text{core}}(U_{\text{core}}, V_{\text{core}}, \{C_{\text{core}}^{(t)}\}_{t=1}^T).$$

**Mutual inverses.** Finally, note that for any feasible $(U_{\text{core}}, V_{\text{core}}, \{C_{\text{core}}^{(t)}\})$,

$$U_B^\top (U_B U_{\text{core}}) = U_{\text{core}}, \qquad V_A^\top (V_A V_{\text{core}}) = V_{\text{core}},$$

so applying the two constructions in succession yields the identity mapping on the core variables. Likewise, for any feasible $(U, V, \{C^{(t)}\})$ in the restricted set with $\text{Col}(U) \subseteq \text{Col}(U_B)$ and $\text{Col}(V) \subseteq \text{Col}(V_A)$, we can write $U = U_B R_U$, $V = V_A R_V$ and then

$$U_B(U_B^\top U) = U_B(U_B^\top U_B R_U) = U_B R_U = U,$$

and similarly for $V$, showing that the constructions are also inverse to each other on the restricted original feasible set.

This proves that the mappings between $(U, V, \{C^{(t)}\})$ and $(U_{\text{core}}, V_{\text{core}}, \{C_{\text{core}}^{(t)}\})$ are mutually inverse and preserve the objective value in both directions, which is exactly the claim of the lemma.

$\square$

***Proof of Theorem 4.3.*** By Theorem 4.2, Problem 2 admits an optimal solution whose factors $(U, V, \{C^{(t)}\})$ satisfy $\text{Col}(U) \subseteq \text{Col}(U_B)$ and $\text{Col}(V) \subseteq \text{Col}(V_A)$. Thus we may restrict Problem 2 to this reduced feasible set $\mathcal{F}_{\text{res}}$ without changing its optimal value.

On this restricted domain, Lemma A.1 establishes a bijection

$$\Phi : \mathcal{F}_{\text{res}} \longleftrightarrow \mathcal{F}_{\text{core}}$$

between $\mathcal{F}_{\text{res}}$ and the feasible set $\mathcal{F}_{\text{core}}$ of Problem 11, with inverse $\Phi^{-1} = \Psi$. And the two objective functions are preserved:

$$\mathcal{L}(U, V, \{C^{(t)}\}) = \mathcal{L}_{\text{core}}\big(\Phi(U, V, \{C^{(t)}\})\big),$$
$$\mathcal{L}_{\text{core}}(U_{\text{core}}, V_{\text{core}}, \{C_{\text{core}}^{(t)}\}) = \mathcal{L}\big(\Phi^{-1}(U_{\text{core}}, V_{\text{core}}, \{C_{\text{core}}^{(t)}\})\big).$$

Therefore, the two problems have the same optimal value, and $(U, V, \{C^{(t)}\}) \in \mathcal{F}_{\text{res}}$ is optimal for Problem 2 if and only if $\Phi(U, V, \{C^{(t)}\})$ is optimal for Problem 11, and vice versa. Unfolding the explicit form of $\Phi$ and $\Phi^{-1}$ stated in Lemma A.1 yields exactly the correspondence claimed in Theorem 4.3.

$\square$

# B. Additional Discussions

## B.1. Difference with Alignment-based Baselines.

Alignment-based methods first map each LoRA update to a shared coordinate system $R^{(t)} = \mathrm{Proj}(\Delta W^{(t)})$, merge them to obtain $R_{\mathrm{merged}}$, and reconstruct $\Delta W_{\mathrm{merged}} = \mathrm{Recons}(R_{\mathrm{merged}})$. KnOTS (Stoica et al., 2025) concatenates full updates and performs SVD to obtain per-task aligned matrices $[\Delta W^{(1)}, \ldots, \Delta W^{(T)}] = U\Sigma[V^{(1)}; \ldots; V^{(T)}]^\top$, then merges $\{V^{(t)}\}$ and reconstructs with the shared $(U, \Sigma)$. CoreSpace (Panariello et al., 2025) uses $\mathrm{Core}^{(t)} = U_B^\top \Delta W^{(t)} V_A$ (Eq. 9) as $R^{(t)}$.

These projections are designed to be information-preserving, i.e., we can have $\Delta W^{(t)} = \mathrm{Recons}(\mathrm{Proj}(\Delta W^{(t)}))$. That's why we can use the lossless CoreSpace coordinates to accelerate solving the CtM subspace-learning objective (Sec. 4.3). However, since the projections preserves all information, they do not impose a rank-$r$ bottleneck, obtaining a single rank-$r$ LoRA typically still requires post-hoc truncation (e.g., Eq. 1). In contrast, CtM learns a lossy projection bottleneck and thus guarantees the rank constraint by construction.

## B.2. Detailed Time Complexity Analysis

### B.2.1. TIME COMPLEXITY OF ORIGINAL PROBLEM

**HOSVD initialization.** Let $\mathcal{X}_{\mathrm{mode}-1} \in \mathbb{R}^{n \times (mT)}$ be the mode-1 unfolding of $\mathcal{X}$. We compute its SVD $\mathcal{X}_{\mathrm{mode}-1} = U_{\mathrm{mode}-1}\Sigma_{\mathrm{mode}-1}V_{\mathrm{mode}-1}^\top$ and set the initial left factor $U_{(0)} := U_{\mathrm{mode}-1,[:,1:r]} \in \mathbb{R}^{n \times r}$, i.e., the matrix formed by the top-$r$ left singular vectors of $\mathcal{X}_{\mathrm{mode}-1}$. Similarly, let $\mathcal{X}_{\mathrm{mode}-2} \in \mathbb{R}^{m \times (nT)}$ be the mode-2 unfolding of $\mathcal{X}$, compute $\mathcal{X}_{\mathrm{mode}-2} = U_{\mathrm{mode}-2}\Sigma_{\mathrm{mode}-2}V_{\mathrm{mode}-2}^\top$, and define the initial right factor $V_{(0)} := U_{\mathrm{mode}-2,[:,1:r]} \in \mathbb{R}^{m \times r}$. Given $U_{(0)}$ and $V_{(0)}$, the corresponding core tensor is obtained in closed form as $\mathcal{G}_{(0)} = \mathcal{X} \times_1 U_{(0)}^\top \times_2 V_{(0)}^\top$.

Suppose $Tr \ll m, Tr \ll n, n = m$. The time complexity of performing SVD on $\mathcal{X}_{\mathrm{mode}-1} \in \mathbb{R}^{n \times (mT)}$ is $\mathcal{O}\left(n^2 mT\right)$, and the time complexity of performing SVD on $\mathcal{X}_{\mathrm{mode}-2} \in \mathbb{R}^{m \times (nT)}$ is $\mathcal{O}\left(m^2 nT\right)$. Given $n = m$, the total time complexity is $\mathcal{O}\left(n^3 T\right)$.

**HOOI refinement.** Starting from $U_{(0)}$ and $V_{(0)}$, HOOI alternately updates $U$ and $V$ while keeping the third-mode factor fixed as $I_T$. At iteration $k$, to update $U$, we fix $V = V_{(k)}$ and define $\mathcal{Y} = \mathcal{X} \times_2 V_{(k)}^\top \in \mathbb{R}^{n \times r \times T}$. Let $\mathcal{Y}_{\mathrm{mode}-1} \in \mathbb{R}^{n \times (rT)}$ be the mode-1 unfolding of $\mathcal{Y}$, compute its SVD $\mathcal{Y}_{\mathrm{mode}-1} = \tilde{U}_{\mathrm{mode}-1}\tilde{\Sigma}_{\mathrm{mode}-1}\tilde{V}_{\mathrm{mode}-1}^\top$, and set $U_{(k+1)} := \tilde{U}_{\mathrm{mode}-1,[:,1:r]}$. Similar operation could be done to update $V$. After updating $U$ and $V$, the core tensor is recomputed as $\mathcal{G}_{(k+1)} = \mathcal{X} \times_1 U_{(k+1)}^\top \times_2 V_{(k+1)}^\top$. This alternating scheme monotonically decreases the objective in Problem 8 and converges to a stationary point of the Tucker-2 approximation problem.

The projection step, i.e., computing $\mathcal{Y} = \mathcal{X} \times_2 V_{(k)}^\top \in \mathbb{R}^{n \times r \times T}$, involves a matrix multiplication between a matrix with shape $n \times m \times T$ and a matrix with shape $m \times r$. The time complexity is $\mathcal{O}\left(nmTr\right)$. The time complexity of performing SVD on $\mathcal{Y}_{\mathrm{mode}-1} \in \mathbb{R}^{n \times (rT)}$ is $\mathcal{O}\left(n(Tr)^2\right)$. Given $Tr \ll m, Tr \ll n$ and $n = m$, the time complexity is dominated by the projection step and is $\mathcal{O}\left(n^2 Tr\right)$. Thus, the total time complexity of HOOI is $\mathcal{O}\left(K \times n^2 Tr\right)$, where $K$ is the iteration number.

### B.2.2. TIME COMPLEXITY OF CORE-SPACE PROBLEM

**Construct the core space and compute $M^{(t)}$.** Constructing the core space involves performing SVD of a matrix with shape $n \times (Tr_{\mathrm{in}})$ and a matrix with shape $(Tr_{\mathrm{in}}) \times m$. The time complexity is $\mathcal{O}\left(n(Tr_{\mathrm{in}})^2 + m(Tr_{\mathrm{in}})^2\right)$. To compute $M^{(t)}$, we can first compute $U_B^\top B^{(t)}$ and $A^{(t)}V_A$, then compute $M^{(t)} = \left(U_B^\top B^{(t)}\right)\left(A^{(t)}V_A\right)$. The time complexity of computing one matrix is $\mathcal{O}\left(nT r_{\mathrm{in}}^2 + mT r_{\mathrm{in}}^2 + r_{\mathrm{in}}(Tr_{\mathrm{in}})^2\right)$. Compute $T$ matrices requires $\mathcal{O}\left(n(Tr_{\mathrm{in}})^2 + m(Tr_{\mathrm{in}})^2 + (Tr_{\mathrm{in}})^3\right)$ Given $Tr \ll n, Tr \ll m, m = n, r = r_{\mathrm{in}}$, the total time complexity is $\mathcal{O}\left(n(Tr)^2\right)$.

**HOSVD initialization.** Similar to the previous analysis, HOSVD initialization needs to perform two SVDs on $(Tr_{\mathrm{in}}) \times (T^2 r_{\mathrm{in}})$ matrices. The complexity is $\mathcal{O}\left((Tr_{\mathrm{in}})^2(T^2 r_{\mathrm{in}})\right)$. Given $r = r_{\mathrm{in}}$, the final complexity is $\mathcal{O}\left(T^4 r^3\right)$.

**HOOI refinement.** Each update step first projects a tensor with shape $(Tr_{\mathrm{in}}) \times (Tr_{\mathrm{in}}) \times T$ into $(Tr_{\mathrm{in}}) \times r \times T$. The time complexity is $\mathcal{O}\left(T^3 r_{\mathrm{in}}^2 r\right)$. The following SVD is performed on a matrix with shape $(Tr_{\mathrm{in}}) \times (rT)$, the time complexity is also $\mathcal{O}\left((Tr)^3\right)$ with $r_{\mathrm{in}} = r$. Thus, the final time complexity is $\mathcal{O}\left((Tr)^3 \times K\right)$, where $K$ is the iteration number.

## B.3. Algorithm

Algorithm 1 gives the algorithm of Compress-then-Merge pipeline with the core-space acceleration.

---

**Algorithm 1** Compress-then-Merge pipeline

---

**input** $T$ LoRA updates $\{(A^{(t)}, B^{(t)})\}_{t=1}^{T}$ with input rank $r_{\text{in}}$, target rank $r$, hyperparameter $\beta \in [0, 1]$, base merging method $\mathcal{M}$.

**output** Merged low-rank update $\Delta W_{\text{LoRA}}$.

1: Stack the LoRA factors respectively and compute the SVD to obtain $U_B$ and $V_A$ (Eq 9):

$$B_{\text{cat}} = \begin{bmatrix} B^{(1)}, \ldots, B^{(T)} \end{bmatrix} = U_B \Sigma_B V_B^{\top} \in \mathbb{R}^{n \times Tr_{\text{in}}},$$
$$A_{\text{cat}} = \begin{bmatrix} A^{(1)}; \ldots; A^{(T)} \end{bmatrix} = U_A \Sigma_A V_A^{\top} \in \mathbb{R}^{Tr_{\text{in}} \times m}.$$

2: Compute $M^{(t)} = (U_B^{\top} B^{(t)})(A^{(t)} V_A)$ and then compute $M_{\text{target}}^{(t)}$ (Eq 3-4):

$$\|M\|_{F,\text{Avg}} = \frac{1}{T} \sum_{t=1}^{T} \|M^{(t)}\|_F$$
$$\lambda^{(t)} = \beta \|M^{(t)}\|_F + (1 - \beta) \|M\|_{F,\text{Avg}},$$
$$M_{\text{target}}^{(t)} = \lambda^{(t)} \frac{M^{(t)}}{\|M^{(t)}\|_F}.$$

3: Solve the following problem to obtain $U_{\text{core}}, V_{\text{core}}$ (Problem 11):

$$\min_{\substack{U_{\text{core}}^{\top} U_{\text{core}} = I_r, V_{\text{core}}^{\top} V_{\text{core}} = I_r, \\ \{C_{\text{core}}^{(t)}\}_{t=1}^{T}}} \sum_{t=1}^{T} \left\| M_{\text{target}}^{(t)} - U_{\text{core}} C_{\text{core}}^{(t)} V_{\text{core}}^{\top} \right\|_F^2,$$

4: Lift back to the original space set $U = U_B U_{\text{core}}$ and $V = V_A V_{\text{core}}$.

5: Compute low-rank coordinates for each task:

$$O^{(t)} = U^{\top} \Delta W^{(t)} V.$$

6: Merge in the shared subspace:

$$O_{\text{merged}} = \mathcal{M}\big(\{O^{(t)}\}_{t=1}^{T}\big).$$

7: Lift back to obtain the low-rank merged result: $\Delta W_{\text{LoRA}} = U\, O_{\text{merged}}\, V^{\top}$.

---

## C. Experimental Details

### C.1. Datasets and Models

Following KnOTS (Stoica et al., 2025) and CoreSpace (Panariello et al., 2025), we evaluate merging methods on both vision tasks and NLI tasks.

**Models.**    For reproducibility, we adopt the publicly released KnOTS LoRA adapters on HuggingFace as our task-specific experts. For vision tasks, we use CLIP ViT-B/32 as the base model, and the LoRAs are trained with rank $r_{\mathrm{in}}$=16, LoRA scaling $\alpha$=16, and dropout 0.1, with no bias parameters trained (`bias=none`). The adapters are injected into the attention projection modules {`q_proj, k_proj, v_proj, out_proj`} wherever they appear in the backbone. A fixed classification head is constructed using the text embeddings of the dataset class labels of the CLIP text encoder (i.e., the classifier weights are obtained by encoding class names with the text encoder). For NLI tasks, we use the LLaMA3-8B as the base model and the LoRAs are trained with the same configuration of vision task (the `out_proj` in LLaMA3-8B is `o_proj`). (Stoica et al., 2025) instantiates the model with a sequence-classification head (3-way NLI) using HuggingFace `AutoModelForSequenceClassification`. During merging, only the LLaMA backbone (with LoRA layers) is merged; evaluation on each dataset uses its corresponding task-specific classification head (for binary NLI datasets, the missing label is masked).

**Datasets.**    For vision task, we adopt the following eight datasets spanning fine-grained recognition, textures, remote sensing, traffic signs, digits, and scene classification.

- **Cars** (Krause et al., 2013): a fine-grained recognition benchmark for distinguishing car categories.
- **Describable Textures Dataset (DTD)** (Cimpoi et al., 2014): a texture classification dataset organized by human-describable texture attributes (e.g., "striped", "dotted").
- **EuroSAT** (Helber et al., 2018): a remote-sensing land-use/land-cover classification dataset built from Sentinel-2 imagery.
- **German Traffic Sign Recognition Benchmark (GTSRB)** (Stallkamp et al., 2012): a traffic-sign recognition benchmark featuring real-world German road signs under diverse imaging conditions.
- **MNIST** (LeCun et al., 1998): a classic handwritten digit classification dataset widely used for benchmarking recognition methods.
- **NWPU-RESISC45** (Cheng et al., 2017): a remote-sensing scene classification dataset with substantial intra-class variation and inter-class similarity.
- **SUN397** (Xiao et al., 2016): a large-scale scene recognition dataset covering a broad range of indoor and outdoor environments.
- **Street View House Numbers (SVHN)** (Netzer et al., 2011): a digit classification benchmark based on house-number images from Google Street View.

For NLI task, we adopt the following six datasets:

- **SNLI** (Bowman et al., 2015): a large NLI dataset of sentence pairs labeled as entailment, neutral, or contradiction, constructed mainly from image captions.
- **MNLI** (Williams et al., 2018): a multi-genre NLI benchmark with the same three-way labels as SNLI, designed to test cross-domain generalization.
- **SICK** (Marelli et al., 2014): a sentence-pair benchmark for textual entailment and semantic relatedness.
- **QNLI** (Wang et al., 2019): a binary GLUE task created by recasting SQuAD (Rajpurkar et al., 2016) question answering into sentence-pair inference (entailment vs. not-entailment).
- **RTE** (Wang et al., 2019): a binary textual entailment task in GLUE assembled from multiple RTE challenge datasets.
- **SCITAIL** (Khot et al., 2018): a science-domain entailment dataset built from multiple-choice QA, pairing retrieved science sentences (premises) with question–answer hypotheses.

### C.2. Implementation Details and Hyperparameter Search

All compared methods are evaluated under the same hyperparameter search protocol. Following the setup of Panariello et al. (2025), we perform a linear search on a holdout validation set and report the corresponding performance on the test set using the best validation configuration.Different methods involve different sets of hyperparameters, which we describe below.

All methods include a scaling factor $\alpha$ that controls the overall magnitude of the merged task vector. Unless otherwise specified, we search $\alpha \in \{0.000, 0.025, 0.050, \ldots, 4.000\}$ with step $0.025$.

TIES introduces one additional hyperparameter, the retention coefficient top-$K$, which controls the fraction of parameters retained during pruning: the largest $K\%$ entries (by absolute magnitude) are preserved. We search $K \in \{0, 5, 10, \ldots, 100\}$ with step $5$.

DARE-TIES uses the same top-$K$ retention coefficient as TIES. In addition, it includes a pruning (drop) rate that controls the strength of random pruning applied before performing the TIES merge. We search this pruning coefficient in $p \in \{0.00, 10^{-5}, 0.05, 0.10, \ldots, 1.00\}$ with step $0.05$. Following the Panariello et al. (2025) codebase, we include $10^{-5}$ as a near-zero value (in addition to 0) to avoid implementation edge cases at exactly zero while remaining effectively "no pruning".

Iso-C does not introduce additional hyperparameters. However, its whitening step normalizes the singular spectrum (i.e., it flattens the singular values), which makes a subsequent truncated SVD ill-defined. Therefore, we first apply truncation to the merged update produced by TA to obtain a rank-$r$ approximation, and then apply the whitening operation to the truncated result.

LoRA-LEGO introduces one additional hyperparameter $k$, which determines the number of clusters/components used in its low-rank merging procedure. Since our goal is to output a single rank-$r$ LoRA adapter, we set $k = r$ in all experiments.

RobustMerge introduces an additional top-$K$ retention coefficient (defined in the same way as in TIES), which we search over $K \in \{0, 5, 10, \ldots, 100\}$.

Iso-CTS explicitly decomposes the merged update into two parts: *common* directions extracted from the summed task matrices and *task-specific* directions extracted from each task's residual matrix. We set the task-specific subspace dimension to 1 per task, and the common subspace dimension to $r - T$, where $r$ is the target rank and $T$ is the number of tasks. The remaining hyperparameter, the scaling factor $\alpha$, is selected via a search following the default procedure described above.

Our method introduces the reweighting hyperparameter $\beta$ and we search it in $\{0.0, 0.25, 0.5, 0.75\}$. For each value, we use the linear search strategy to search other hyperparameters.

The searched optimal parameters used in Table 1 and Table 2 are provided in Table 5 and Table 6, respectively. Importantly, the best hyperparameters for the pre-compression merge and the post-compression are searched independently, so $\mathrm{Avg}_{\mathrm{full}}$ and $\mathrm{Avg}$ each reflect their respective tuned optimum.

*Table 5.* Searched optimal hyperparameters for Table 1.

| Method | Space | SVD Truncation | Scaling factor | Top-$K$ | Pruning Ratio | $\beta$ |
|---|---|---|---|---|---|---|
| LoRA-LEGO | – | – | 0.125 | – | – | – |
| RobustMerge | – | – | 1.350 | 30 | – | – |
| Iso-CTS | – | – | 0.250 | – | – | – |
| Iso-C | Full | ✓ | 0.225 | – | – | – |
| | KnOTS | ✓ | 0.125 | – | – | – |
| | CoreSpace | ✓ | 0.225 | – | – | – |
| | Full | ✗ | 0.875 | – | – | – |
| | KnOTS | ✗ | 0.150 | – | – | – |
| | CoreSpace | ✗ | 0.875 | – | – | – |
| TIES | Full | ✓ | 0.325 | 35 | – | – |
| | KnOTS | ✓ | 0.225 | 30 | – | – |
| | CoreSpace | ✓ | 0.675 | 5 | – | – |
| | Full | ✗ | 0.325 | 25 | – | – |
| | KnOTS | ✗ | 0.225 | 25 | – | – |
| | CoreSpace | ✗ | 0.675 | 25 | – | – |
| DARE-TIES | Full | ✓ | 0.300 | 35 | 0.20 | – |
| | KnOTS | ✓ | 0.225 | 30 | 0.00 | – |
| | CoreSpace | ✓ | 0.600 | 10 | 0.10 | – |
| | Full | ✗ | 0.325 | 30 | 0.20 | – |
| | KnOTS | ✗ | 0.200 | 25 | 0.05 | – |
| | CoreSpace | ✗ | 0.600 | 15 | 0.10 | – |
| Iso-C | – | – | 0.225 | – | – | 0.00 |
| TIES | – | – | 0.750 | 25 | – | 0.25 |
| DARE-TIES | – | – | 0.750 | 20 | 0.05 | 0.25 |

*Table 6.* Searched optimal hyperparameters for Table 2.

| Method | Space | SVD Truncation | Scaling factor | Top-$K$ | Pruning Ratio | $\beta$ |
|---|---|---|---|---|---|---|
| LoRA-LEGO | – | – | 0.425 | – | – | – |
| RobustMerge | – | – | 0.825 | 30 | – | – |
| Iso-CTS | – | – | 0.850 | – | – | – |
| | Full | ✓ | 0.825 | – | – | – |
| | KnOTS | ✓ | 0.500 | – | – | – |
| Iso-C | CoreSpace | ✓ | 0.825 | – | – | – |
| | Full | ✗ | 2.575 | – | – | – |
| | KnOTS | ✗ | 0.600 | – | – | – |
| | CoreSpace | ✗ | 2.850 | – | – | – |
| | Full | ✓ | 0.725 | 10 | – | – |
| | KnOTS | ✓ | 0.550 | 30 | – | – |
| TIES | CoreSpace | ✓ | 1.100 | 80 | – | – |
| | Full | ✗ | 0.750 | 10 | – | – |
| | KnOTS | ✗ | 0.575 | 35 | – | – |
| | CoreSpace | ✗ | 1.075 | 75 | – | – |
| | Full | ✓ | 0.725 | 10 | 1e-5 | – |
| | KnOTS | ✓ | 0.550 | 30 | 0.25 | – |
| DARE-TIES | CoreSpace | ✓ | 1.000 | 30 | 1e-5 | – |
| | Full | ✗ | 0.725 | 10 | 0.10 | – |
| | KnOTS | ✗ | 0.500 | 30 | 0.20 | – |
| | CoreSpace | ✗ | 0.950 | 15 | 1e-5 | – |
| Iso-C | – | – | 0.875 | – | – | 0.25 |
| TIES | – | – | 1.225 | 45 | – | 0.75 |
| DARE-TIES | – | – | 1.225 | 55 | 0.10 | 0.75 |

# D. Additional Experimental Results

## D.1. CtM Learns a More Effective and Balanced Subspace from Both Energy and Functional Perspective.

To compare the subspaces learned by CtM and MtC, we evaluate them from both the energy and functional perspectives. For the energy part, we measure how much of the original LoRA updates can be captured by the left/right subspaces induced by their rank-$r$ merged results. For the function part, we further evaluate how much task utility the projected LoRAs can preserve compared to that of the original task-specific LoRAs.

Assume there are $N$ LoRA-injected layers. Consider any method that outputs a rank-$r$ update $\Delta W^i_{\mathrm{LoRA}}$ at layer $i$. We extract its left/right subspaces $(U^i, V^i)$ via the thin SVD, $\Delta W^i_{\mathrm{LoRA}} = U^i \Sigma^i V^{i\top}$, where $U^{i\top} U^i = I_r$ and $V^{i\top} V^i = I_r$. Given the original task update $\Delta W^{(t),i}$ at layer $i$, we project it onto $(U^i, V^i)$ by

$$\widehat{\Delta W}^{(t),i} = U^i U^{i\top} \Delta W^{(t),i} V^i V^{i\top}.$$

We define the per-layer energy retention ratio as

$$\rho^{(t),i} = \frac{\left\| \widehat{\Delta W}^{(t),i} \right\|_F}{\left\| \Delta W^{(t),i} \right\|_F} \in [0, 1].$$

We report the layer-averaged retention

$$\overline{\rho}^{(t)} = \frac{1}{N} \sum_{i=1}^{N} \rho^{(t),i}.$$

A higher $\overline{\rho}^{(t)}$ indicates a subspace that better captures the original LoRA updates across layers.

For the functional evaluation, we compute the accuracy of the original LoRA on its corresponding task, denoted as $\mathrm{Acc}_{\mathrm{ori}}$, and the accuracy of the projected LoRA, denoted as $\mathrm{Acc}_{\mathrm{proj}}$. We then define the accuracy retention as

$$\eta^{(t)} = \frac{\mathrm{Acc}_{\mathrm{proj}}}{\mathrm{Acc}_{\mathrm{ori}}}.$$

This metric directly measures how much task utility is preserved after projecting the original task LoRA onto the shared rank-$r$ subspace.

For CtM, $\Delta W^i_{\mathrm{LoRA}}$ is the rank-$r$ update $U O_{\mathrm{merged}} V^\top$. For MtC, we use *task arithmetic*, which linearly sums the LoRA updates. This avoids non-linear merging methods (e.g., TIES) that yield updates outside the linear span of the original LoRAs.

Figure 3 summarizes the results. CtM consistently achieves higher or similar energy retention after projection, indicating that its rank-$r$ subspace captures a larger fraction of the original LoRA updates. More importantly, CtM exhibits substantially lower variance across datasets, suggesting a more balanced and stable coverage of task directions. In contrast, MtC shows significant collapse on several tasks, despite comparable coverage on others, reflecting its sensitivity to scale and truncation.

The functional evaluation leads to a consistent conclusion. CtM improves the mean accuracy retention, and raises the worst-task retention significantly. This shows that CtM does not merely preserve more parameter-space energy; it also retains the task-relevant components needed for downstream performance. Overall, the energy and functional results jointly demonstrate that CtM learns a subspace that better preserves multi-task structure and task utility under a fixed rank constraint.

Beyond the CLIP results, we further conduct the same functional projection test on NLI, as reported in Figure 6. The conclusion remains consistent from the perspectives of mean performance, variance, and worst-case robustness: CtM improves the mean functional retention from $93.24\%$ to $96.43\%$, reduces the cross-task standard deviation, and, more importantly, substantially improves the worst-task retention from $66.86\%$ to $91.13\%$. Specifically, MtC suffers a sharp collapse on SICK, which both lowers its mean retention and increases its cross-task variability. In contrast, CtM avoids such severe task collapse and produces a more stable retention profile across tasks. This result shows that the subspace learned by CtM is more balanced under the fixed-rank constraint, improving the average retained utility and providing stronger protection against worst-case task degradation.

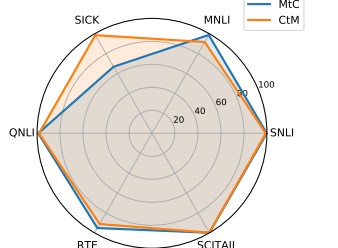

| | MtC | CtM |
|---|---|---|
| Mean | 93.24 | 96.43 |
| Worst | 66.86 | 91.13 |
| Std | 13.04 | 3.91 |

*Figure 6.* Function (accuracy) retention of CtM and MtC on NLI.

### D.2. Core-space Acceleration with Preserved Solution Quality.

Table 4 shows that optimizing in the core space is substantially faster, while preserving the final merged performance.

To quantify how close the shared subspaces learned in the full space and in the core space are, we use two distances: a basis-invariant subspace distance (Chordal) and a basis-aligned element-wise distance. Suppose each adapter contains $N$ LoRA-injected layers. For layer $i$, let $(U_{\text{full}}^i, V_{\text{full}}^i)$ denote the rank-$r$ left/right subspaces learned by optimizing Problem 2 directly in the full space, and let $(U^i, V^i)$ denote the corresponding rank-$r$ subspaces obtained from the core-space formulation (i.e., after lifting the core-space solution back to the full space). We compute a per-layer distance for the left and right subspaces, denoted as $\text{Dis}_U^i$ and $\text{Dis}_V^i$, and report the average distance across all layers:

$$\text{AvgDis} = \frac{1}{2N} \sum_{i=1}^{N} \left( \text{Dis}_U^i + \text{Dis}_V^i \right).$$

**Chordal distance.** Chordal distance depends only on the subspaces (their column spaces) and is invariant to any rotation of the chosen bases. Concretely, given two semi-orthogonal matrices $Q_1, Q_2 \in \mathbb{R}^{d \times r}$ with $Q_1^\top Q_1 = Q_2^\top Q_2 = I_r$, the chordal distance is

$$d_{\text{chord}}(Q_1, Q_2) = \left\| \sin \Theta \right\|_F = \sqrt{r - \|Q_1^\top Q_2\|_F^2},$$

where $\Theta = \text{diag}(\theta_1, \dots, \theta_r)$ contains the principal angles between the two subspaces. For $r$-dimensional subspaces, $d_{\text{chord}}(\cdot, \cdot) \in [0, \sqrt{r}]$: 0 indicates identical subspaces, while $\sqrt{r}$ corresponds to maximally separated (orthogonal) subspaces. We set $\text{Dis}_U^i = d_{\text{chord}}(U_{\text{full}}^i, U^i)$ and $\text{Dis}_V^i = d_{\text{chord}}(V_{\text{full}}^i, V^i)$ for each layer $i$.

**Element-wise distance.** While the chordal distance compares subspaces and is invariant to any rotation of the chosen bases, the coordinates used for merging depend on the specific bases $(U, V)$. A rotation within the same $r$-dimensional subspace can change coordinates substantially: for instance, in $\mathbb{R}^2$, the coordinate $(1, 0)$ becomes $\left(\frac{\sqrt{2}}{2}, \frac{\sqrt{2}}{2}\right)$ after a $\frac{\pi}{4}$ rotation. This matters because non-linear merging methods such as TIES operate element-wise (e.g., through sign-based masking and pruning) and are therefore sensitive to basis rotations, even when the underlying subspaces are identical.

To more strictly assess whether the bases learned in the full space and the core space match, we also report an element-wise absolute distance between the corresponding semi-orthogonal matrices. Given $Q_1, Q_2 \in \mathbb{R}^{d \times r}$ with orthonormal columns, we define

$$d_{\text{elem}}(Q_1, Q_2) = \frac{1}{dr} \left\| Q_1 - Q_2 \right\|_1,$$

where $\| \cdot \|_1$ denotes the entry-wise $\ell_1$ norm (sum of absolute values). We set $\text{Dis}_U^i = d_{\text{elem}}(U_{\text{full}}^i, U^i)$ and $\text{Dis}_V^i = d_{\text{elem}}(V_{\text{full}}^i, V^i)$ for each layer $i$.

To avoid possible sign ambiguity of basis vectors (a sign flip would substantially change the element-wise distance but would not affect TIES), we make $d_{\text{elem}}$ invariant to column-wise sign flips by fixing a deterministic sign convention. For each column $j$, we choose an index $p = \arg\max_i |Q_{ij}|$, and if $Q_{pj} < 0$ we multiply the entire column by $-1$. We apply this canonicalization independently to $Q_1$ and $Q_2$, and then compute $d_{\text{elem}}(Q_1, Q_2)$.

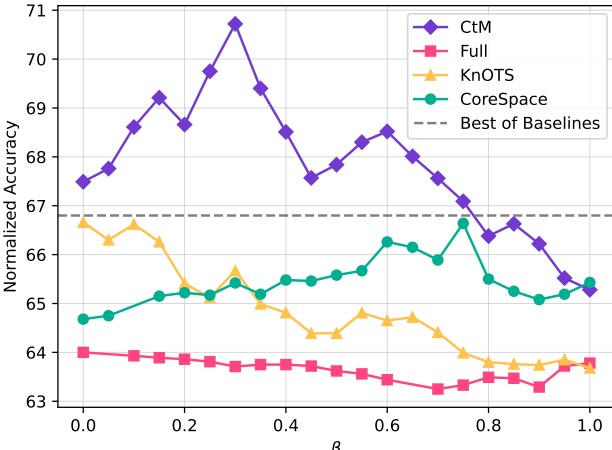

*Figure 7.* Normalized accuracy when applying $\beta$-reweighting to MtC baselines. All methods are evaluated using TIES as the base merging method. Reweighting improves the best baseline modestly, but CtM remains substantially higher for small-to-moderate $\beta$, showing that CtM's gains are not attributable to an extra hyperparameter alone.

### D.3. Applying $\beta$-Reweighting to MtC Baselines.

**How $\beta$ is used in CtM.** CtM introduces a single hyperparameter $\beta$ through the task-importance weights $\lambda^{(t)}$ (Eq. 3–4). Importantly, this reweighting is applied *only* when learning the shared subspace: $(U, V)$ is learned by fitting the surrogate targets $\Delta W_{\text{target}}^{(t)}$, while the merge-time coordinates are computed from the *original* updates,

$$O^{(t)} = U^\top \Delta W^{(t)} V.$$

Therefore, $\beta$ controls which directions are preserved under the rank-$r$ bottleneck, rather than directly rescaling the quantities being merged.

Although projection onto a learned rank-$r$ subspace is generally lossy and may change the per-task Frobenius norm ($\|O^{(t)}\|_F$ versus $\|\Delta W^{(t)}\|_F$), our subspace-coverage analysis in Section 5.3 demonstrates that the *energy retention ratio* remains similar across tasks. This indicates that the projection step tends to reduce task magnitudes in a roughly proportional manner, and does not act as an additional, task-specific reweighting at merge time.

**$\beta$-transfer as a control test.** We further investigate whether $\beta$-reweighting acts as a generic scale-compensation mechanism that could account for CtM's performance gains. To this end, we introduce the same degree of freedom into the MtC baselines: for a given $\beta$, we rescale each task's update according to Eq. 3 (i.e., using the same $\lambda^{(t)}$ rule). We then apply the unchanged original MtC pipeline on the rescaled updates. All experiments with different $\beta$ values are conducted by re-searching the hyperparameters to ensure fairness.

Figure 7 shows that allowing $\beta$ yields very limited improvements for the strongest baseline (notably CoreSpace), consistent with the intuition that task reweighting can partially mitigate scale-dominance effects. However, even after sweeping $\beta$, the best baseline remains far below CtM. This suggests that CtM's advantage is not primarily driven by possessing an additional tunable parameter. Instead, it comes from enforcing the rank constraint *before* merging by learning a shared subspace and performing merging directly in the resulting low-dimensional coordinates, thereby avoiding the brittle post-hoc truncation step inherent to MtC.

Overall, these results support the following takeaway: reweighting alone provides limited gains for MtC, whereas in CtM $\beta$ interacts with the rank constraint to *shape* the shared subspace, leading to substantially better multi-task trade-offs.

### D.4. CtM with Additional Base Merge Rules

Beyond the main results in Tables 1-2, we additionally combine CtM with a broader set of training-free merge rules, including Task Arithmetic (Ilharco et al., 2023), TSV (Gargiulo et al., 2025), and PCB-Merging (Du et al., 2024). The detailed results

are reported in Table 7. Overall, CtM either improves or matches the corresponding MtC baselines. Together with the main experiments, we observe that the largest gains typically occur for *coordinate-wise* rules (e.g., TIES, DARE-TIES, and PCB-Merging). We hypothesize that this is because, after CtM projects task updates into a compact $r \times r$ coordinate system, the cross-task conflicts are concentrated and are easier for coordinate-wise heuristics to detect and resolve.

We note that for TSV, CtM yields almost no improvement over its MtC counterpart. We hypothesize that this behavior stems from TSV's explicit *per-task spectral budget* in the merge coordinate system. Concretely, TSV allocates each task roughly a $1/T$ fraction of the available singular directions. When TSV operates on a high-dimensional coordinate space (e.g., an $n \times n$ matrix), this corresponds to an effective per-task budget of about $n/T$ directions. In contrast, under CtM all task updates are already constrained to lie in a shared $r \times r$ coordinate space, so the same $1/T$ budgeting reduces the effective per-task capacity to approximately $r/T$ directions. For our typical setting with $r{=}16$ and $T{=}8$, this leaves each task with only about two singular directions (i.e., a rank-2 budget), which can be overly restrictive and may discard substantial task-relevant information. As a result, CtM offers limited headroom for TSV and the two pipelines perform similarly.

Finally, although the magnitude of gains varies across merge rules, these results consistently suggest that enforcing the rank-$r$ bottleneck *before* merging is a robust design choice: CtM rarely hurts and often improves upon MtC. At the same time, the varying gains across different merge rules suggest that some merging rules can better exploit the shared $r \times r$ coordinates than others. This gap indicates room for developing merge rules that are more compatible with CtM, i.e., explicitly designed to mitigate conflicts *within* the shared low-dimensional coordinate space. We leave the exploration of such CtM-tailored merging operators to future work.

*Table 7.* Normalized accuracies on 8 vision tasks with ViT-B/32 with more base merging rules.

| Method | Space | Cars | DTD | EuroSAT | GTSRB | MNIST | RESISC | SUN397 | SVHN | Avg |
|---|---|---|---|---|---|---|---|---|---|---|
| *LoRA's absolute accuracy* | | *74.00* | *58.30* | *99.00* | *92.70* | *99.30* | *88.40* | *64.50* | *96.20* | – |
| Task Arithmetic | Full | 81.82 | 72.81 | 47.51 | 38.98 | 52.30 | 69.87 | 96.92 | 40.67 | 62.61 |
| | KnOTS | 81.65 | 72.63 | 47.66 | 38.97 | 52.44 | 70.03 | 96.92 | 40.64 | 62.62 |
| | CoreSpace | 81.76 | 72.81 | 47.51 | 39.00 | 52.44 | 69.92 | 96.92 | 40.70 | 62.63 |
| | CtM (ours) | 81.04 | 73.81 | 49.08 | 39.96 | 51.32 | 70.76 | 97.57 | 39.00 | 62.82 |
| TSV | Full | 83.40 | 74.08 | 49.38 | 37.76 | 55.97 | 70.78 | 97.30 | 40.29 | 63.62 |
| | KnOTS | 82.24 | 73.45 | 49.61 | 38.65 | 53.50 | 70.01 | 97.43 | 39.84 | 63.09 |
| | CoreSpace | 83.65 | 72.81 | 48.97 | 36.84 | 56.78 | 70.55 | 97.04 | 42.25 | 63.61 |
| | CtM (ours) | 81.69 | 74.54 | 51.52 | 42.04 | 49.71 | 71.82 | 97.44 | 40.37 | 63.64 |
| PCB-Merging | Full | 82.05 | 72.90 | 50.58 | 37.11 | 53.86 | 69.47 | 97.04 | 40.42 | 62.93 |
| | KnOTS | 82.30 | 72.99 | 46.54 | 38.72 | 54.91 | 69.35 | 96.37 | 43.23 | 63.05 |
| | CoreSpace | 81.76 | 73.72 | 49.91 | 41.05 | 55.56 | 69.36 | 97.45 | 37.71 | 63.31 |
| | CtM (ours) | 81.27 | 72.90 | 60.91 | 37.50 | 61.18 | 69.40 | 96.72 | 50.83 | 66.34 |

## D.5. Detailed Pre-compression Results

Tables 8 and 9 report detailed pre-truncation full-weight results ($\mathrm{Avg}_{\mathrm{full}}$ in Table 1 and Table 2) for MtC baselines.

*Table 8.* Normalized accuracies on 8 vision tasks with ViT-B/32, including full-weight results before SVD truncation (supplementary to Table 1).

| Method | Space | SVD Truncation | Cars | DTD | EuroSAT | GTSRB | MNIST | RESISC | SUN397 | SVHN | Avg |
|---|---|---|---|---|---|---|---|---|---|---|---|
| *LoRA's absolute accuracy* | | | *74.00* | *58.30* | *99.00* | *92.70* | *99.30* | *88.40* | *64.50* | *96.20* | *–* |
| LoRA-LEGO | – | – | 81.42 | 72.72 | 47.03 | 38.89 | 50.88 | 69.83 | 97.21 | 40.90 | 62.36 |
| RobustMerge | – | – | 81.38 | 74.91 | 52.67 | 44.71 | 55.15 | 69.56 | 97.17 | 38.94 | 64.31 |
| Iso-CTS | – | – | 81.93 | 76.91 | 42.99 | 53.01 | 59.42 | 74.10 | 98.35 | 43.98 | 66.34 |
| Iso-C | Full | ✓ | 82.98 | 74.18 | 49.42 | 42.96 | 62.95 | 71.18 | 97.99 | 44.14 | 65.72 |
| | Full | ✗ | 83.42 | 84.30 | 49.42 | 82.08 | 71.41 | 83.39 | 99.01 | 54.23 | 75.91 |
| | KnOTS | ✓ | 80.37 | 73.81 | 46.35 | 38.76 | 48.94 | 69.83 | 97.32 | 37.17 | 61.57 |
| | KnOTS | ✗ | 82.49 | 73.54 | 49.49 | 44.37 | 54.22 | 71.99 | 97.05 | 43.77 | 64.61 |
| | CoreSpace | ✓ | 82.98 | 74.08 | 49.31 | 42.92 | 63.10 | 71.12 | 97.99 | 44.17 | 65.71 |
| | CoreSpace | ✗ | 83.33 | 84.21 | 49.61 | 82.05 | 71.32 | 83.35 | 98.95 | 54.17 | 75.87 |
| TIES | Full | ✓ | 82.51 | 72.90 | 49.94 | 36.93 | 55.41 | 69.22 | 96.95 | 43.15 | 63.38 |
| | Full | ✗ | 82.37 | 72.35 | 51.07 | 37.23 | 57.40 | 69.58 | 96.76 | 44.78 | 63.94 |
| | KnOTS | ✓ | 82.35 | 72.99 | 46.61 | 38.66 | 56.50 | 69.27 | 96.44 | 44.22 | 63.38 |
| | KnOTS | ✗ | 82.64 | 72.72 | 45.79 | 42.09 | 58.41 | 70.37 | 96.21 | 47.01 | 64.40 |
| | CoreSpace | ✓ | 83.54 | 73.81 | 53.09 | 39.97 | 65.18 | 68.05 | 95.25 | 43.73 | 65.33 |
| | CoreSpace | ✗ | 83.84 | 76.00 | 50.47 | 50.65 | 65.22 | 71.28 | 96.50 | 47.85 | 67.73 |
| DARE-TIES | Full | ✓ | 82.51 | 72.81 | 49.57 | 37.03 | 56.12 | 69.31 | 96.90 | 42.75 | 63.37 |
| | Full | ✗ | 82.77 | 72.35 | 50.62 | 36.92 | 58.01 | 69.40 | 96.53 | 45.65 | 64.03 |
| | KnOTS | ✓ | 82.35 | 72.99 | 46.61 | 38.66 | 56.50 | 69.27 | 96.44 | 44.22 | 63.38 |
| | KnOTS | ✗ | 82.85 | 73.72 | 47.25 | 41.72 | 58.16 | 70.71 | 96.75 | 45.20 | 64.55 |
| | CoreSpace | ✓ | 84.01 | 73.45 | 53.95 | 40.90 | 64.55 | 69.13 | 96.17 | 45.10 | 65.91 |
| | CoreSpace | ✗ | 84.38 | 76.27 | 57.28 | 51.28 | 67.21 | 71.79 | 96.46 | 51.72 | 69.55 |
| Iso-C | Ours | – | 80.83 | 77.55 | 44.37 | 56.45 | 56.03 | 73.69 | 97.72 | 43.55 | 66.27 |
| TIES | Ours | – | 82.87 | 75.27 | 64.46 | 38.50 | 78.22 | 70.67 | 97.07 | 50.93 | 69.75 |
| DARE-TIES | Ours | – | 83.04 | 75.00 | 64.95 | 38.28 | 79.98 | 69.90 | 96.53 | 57.73 | 70.68 |

*Table 9.* Normalized accuracies on 6 NLI tasks with LLaMA3-8B, including full-weight results before SVD truncation (supplementary to Table 2).

| Method | Space | SVD Truncation | SNLI | MNLI | SICK | QNLI | RTE | SCITAIL | Avg |
|---|---|---|---|---|---|---|---|---|---|
| *LoRA's absolute accuracy* | | | *92.50* | *90.31* | *91.58* | *94.49* | *89.86* | *96.52* | *–* |
| LoRA-LEGO | – | – | 87.25 | 92.84 | 87.47 | 58.51 | 99.19 | 95.81 | 86.84 |
| RobustMerge | – | – | 89.14 | 92.13 | 89.56 | 60.98 | 100.00 | 96.20 | 88.00 |
| Iso-CTS | – | – | 90.91 | 87.83 | 85.02 | 68.76 | 100.81 | 95.96 | 88.21 |
| Iso-C | Full | ✓ | 90.91 | 90.06 | 83.51 | 67.60 | 100.81 | 96.64 | 88.25 |
| | Full | ✗ | 91.59 | 92.00 | 87.51 | 79.00 | 100.00 | 98.20 | 91.38 |
| | KnOTS | ✓ | 86.75 | 90.07 | 89.12 | 56.48 | 100.00 | 95.57 | 86.33 |
| | KnOTS | ✗ | 94.20 | 94.29 | 91.03 | 71.93 | 102.42 | 98.10 | 91.99 |
| | CoreSpace | ✓ | 90.90 | 90.03 | 83.49 | 67.57 | 100.81 | 96.69 | 88.25 |
| | CoreSpace | ✗ | 91.54 | 90.24 | 87.85 | 76.12 | 99.19 | 97.37 | 90.38 |
| TIES | Full | ✓ | 94.78 | 96.16 | 83.42 | 74.04 | 99.19 | 96.69 | 90.71 |
| | Full | ✗ | 94.85 | 96.50 | 82.55 | 75.44 | 99.19 | 95.86 | 90.73 |
| | KnOTS | ✓ | 91.66 | 95.05 | 90.34 | 80.43 | 101.61 | 97.32 | 92.74 |
| | KnOTS | ✗ | 91.57 | 95.18 | 90.92 | 81.62 | 100.81 | 97.32 | 92.90 |
| | CoreSpace | ✓ | 91.65 | 93.15 | 93.46 | 83.46 | 99.19 | 97.71 | 93.10 |
| | CoreSpace | ✗ | 92.21 | 93.36 | 94.19 | 83.65 | 99.19 | 97.81 | 93.40 |
| DARE-TIES | Full | ✓ | 94.76 | 96.18 | 83.44 | 74.11 | 99.19 | 96.64 | 90.72 |
| | Full | ✗ | 94.62 | 96.60 | 82.13 | 75.39 | 97.58 | 95.81 | 90.36 |
| | KnOTS | ✓ | 92.10 | 95.50 | 91.30 | 81.71 | 100.81 | 97.22 | 93.11 |
| | KnOTS | ✗ | 92.92 | 94.70 | 93.52 | 80.48 | 101.61 | 97.76 | 93.50 |
| | CoreSpace | ✓ | 91.90 | 91.82 | 95.39 | 80.79 | 100.00 | 97.37 | 92.88 |
| | CoreSpace | ✗ | 92.12 | 91.99 | 96.64 | 80.31 | 100.00 | 97.66 | 93.12 |
| Iso-C | Ours | – | 90.73 | 89.72 | 88.07 | 67.67 | 100.81 | 96.0 | 88.83 |
| TIES | Ours | – | 91.08 | 90.26 | 96.11 | 89.71 | 100.81 | 96.93 | 94.15 |
| DARE-TIES | Ours | – | 92.96 | 94.12 | 97.44 | 94.24 | 97.58 | 97.42 | 95.63 |

## D.6. Results with Heterogeneous Settings.

Our main experiments focus on *homogeneous* LoRA adapters, where all input LoRAs share the same rank and injected modules. However, CtM can naturally extend to the *heterogeneous* settings, in which different tasks may adopt different input ranks or inject LoRA to different modules.

**Heterogeneous ranks.** Suppose the $t$-th LoRA has rank $r_{\text{in}}^{(t)}$. In the core-space construction, we concatenate the left and right LoRA factors across tasks as

$$B_{\text{cat}} = \left[B^{(1)}, \ldots, B^{(T)}\right] = U_B \Sigma_B V_B^\top \in \mathbb{R}^{n \times \sum_{t=1}^{T} r_{\text{in}}^{(t)}},$$

$$A_{\text{cat}} = \begin{bmatrix} A^{(1)} \\ A^{(2)} \\ \vdots \\ A^{(T)} \end{bmatrix} = U_A \Sigma_A V_A^\top \in \mathbb{R}^{\sum_{t=1}^{T} r_{\text{in}}^{(t)} \times m}.$$

Let $r_{\text{total}} = \sum_{t=1}^{T} r_{\text{in}}^{(t)}$. From the thin SVDs above, we obtain $U_B \in \mathbb{R}^{n \times r_{\text{total}}}$ and $V_A \in \mathbb{R}^{m \times r_{\text{total}}}$, and represent each task update in the core space by $M^{(t)} = U_B^\top \Delta W^{(t)} V_A \in \mathbb{R}^{r_{\text{total}} \times r_{\text{total}}}$. We then compute the corresponding surrogate targets $M_{\text{target}}^{(t)}$ (following Eq. 3–4) and solve Problem 11 in the $r_{\text{total}} \times r_{\text{total}}$ core space to obtain $U_{\text{core}}, V_{\text{core}} \in \mathbb{R}^{r_{\text{total}} \times r}$. Finally, we lift the solution back to the original parameter space via

$$U = U_B U_{\text{core}} \in \mathbb{R}^{n \times r}, \qquad V = V_A V_{\text{core}} \in \mathbb{R}^{m \times r}.$$

Since CtM merges via projections of the weight updates $\{\Delta W^{(t)}\}$, it does not require any assumption that the input adapters share the same rank. We report results with heterogeneous input ranks in Table 10. RobustMerge does not support heterogeneous-rank inputs and is therefore omitted. The results show that CtM yields the highest average performance, consistent with the homogeneous-rank results.

*Table 10.* Normalized accuracies on 8 vision tasks with ViT-B/32 under heterogeneous input ranks. Each task LoRA uses a different rank, while all methods are evaluated under the same target merged rank $r = 16$.

| Method | Space | Cars | DTD | EuroSAT | GTSRB | MNIST | RESISC | SUN397 | SVHN | Avg |
|---|---|---|---|---|---|---|---|---|---|---|
| *Each LoRA's rank* | | *16* | *16* | *8* | *8* | *4* | *4* | *32* | *32* | – |
| *LoRA's absolute accuracy* | | *74.00* | *58.30* | *98.11* | *96.76* | *98.45* | *91.14* | *68.77* | *96.07* | – |
| LoRA-LEGO | – | 79.01 | 72.26 | 58.51 | 60.22 | 72.52 | 65.99 | 91.75 | 44.98 | 68.16 |
| Iso-CTS | – | 81.57 | 80.47 | 39.22 | 40.20 | 64.40 | 63.10 | 91.76 | 42.63 | 62.92 |
| Iso-C | Full | 80.01 | 72.26 | 54.85 | 54.15 | 83.95 | 65.08 | 92.92 | 61.58 | 70.60 |
| | KnOTS | 79.07 | 72.90 | 40.88 | 33.55 | 92.37 | 65.53 | 91.93 | 40.52 | 64.59 |
| | CoreSpace | 80.01 | 72.17 | 54.62 | 54.07 | 83.96 | 65.15 | 92.93 | 61.59 | 70.56 |
| | CtM (ours) | 80.60 | 72.81 | 56.59 | 55.61 | 82.16 | 65.12 | 92.80 | 62.07 | 70.97 |
| TIES | Full | 79.05 | 71.35 | 55.91 | 47.21 | 68.91 | 66.37 | 91.48 | 49.00 | 66.16 |
| | KnOTS | 79.72 | 72.26 | 59.72 | 50.52 | 65.93 | 64.19 | 90.38 | 46.93 | 66.21 |
| | CoreSpace | 77.68 | 72.35 | 61.16 | 48.85 | 89.30 | 65.48 | 92.10 | 58.74 | 70.71 |
| | CtM (ours) | 78.27 | 70.89 | 67.91 | 51.96 | 93.58 | 66.56 | 91.88 | 64.85 | 73.24 |
| DARE-TIES | Full | 79.47 | 71.53 | 55.23 | 48.19 | 69.41 | 66.82 | 91.75 | 47.81 | 66.28 |
| | KnOTS | 80.79 | 72.90 | 56.02 | 47.12 | 67.52 | 66.96 | 91.42 | 48.47 | 66.40 |
| | CoreSpace | 79.09 | 74.72 | 60.97 | 54.95 | 86.83 | 66.39 | 92.58 | 56.05 | 71.45 |
| | CtM (ours) | 78.44 | 69.71 | 68.29 | 53.71 | 92.32 | 66.74 | 91.71 | 63.14 | 73.01 |

**Heterogeneous modules.** Handling the heterogeneous-module setting is straightforward. For a given layer, we simply skip the LoRAs that are not injected at that layer and only process the remaining ones. Based on our original ViT setup, we fine-tune four LoRAs with modules injected into {`q`, `v`, `mlp.fc1`, `mlp.fc2`} on the corresponding datasets (EuroSAT, MNIST, RESISC, and SVHN), while keeping the other four LoRAs on the standard {`q`, `k`, `v`, `o`} modules.

For layers that are not present in all adapters (e.g., `mlp.fc1`, `mlp.fc2`, `k`, and `o`), we merge only the updates from the adapters that actually contain that layer. The results in Table 11 show that CtM consistently outperforms the corresponding MtC baselines under module heterogeneity. This extends the validation beyond the strictly homogeneous module setting.

*Table 11.* Normalized accuracies on 8 vision tasks with ViT-B/32 under heterogeneous injected modules.

| Method | Space | Cars | DTD | EuroSAT | GTSRB | MNIST | RESISC | SUN397 | SVHN | Avg |
|---|---|---|---|---|---|---|---|---|---|---|
| *LoRA's absolute accuracy* | | *74.00* | *58.30* | *97.81* | *92.70* | *98.99* | *92.05* | *64.50* | *95.76* | *–* |
| Iso-C | Full | 76.76 | 70.16 | 52.71 | 37.47 | 81.60 | 75.17 | 94.89 | 66.55 | 69.41 |
| | KnOTS | 79.30 | 73.26 | 52.33 | 38.01 | 55.78 | 70.39 | 97.42 | 40.04 | 63.32 |
| | CoreSpace | 76.76 | 70.07 | 52.71 | 37.39 | 81.74 | 75.17 | 94.95 | 66.62 | 69.42 |
| | CtM (ours) | 78.88 | 77.00 | 52.03 | 55.22 | 74.92 | 68.20 | 97.70 | 52.68 | 69.58 |
| TIES | Full | 79.22 | 71.07 | 43.77 | 37.28 | 61.90 | 73.25 | 96.32 | 51.92 | 64.34 |
| | KnOTS | 79.13 | 71.99 | 47.78 | 35.97 | 63.68 | 72.39 | 96.32 | 50.81 | 64.76 |
| | CoreSpace | 78.06 | 70.80 | 48.20 | 35.63 | 72.82 | 74.39 | 96.26 | 59.41 | 66.95 |
| | CtM (ours) | 76.42 | 69.25 | 50.59 | 35.16 | 75.79 | 75.58 | 95.99 | 75.60 | 69.30 |
| DARE-TIES | Full | 78.90 | 70.98 | 43.77 | 37.14 | 62.47 | 73.19 | 96.25 | 51.93 | 64.33 |
| | KnOTS | 78.63 | 71.99 | 48.92 | 35.20 | 64.57 | 72.46 | 96.25 | 51.17 | 64.90 |
| | CoreSpace | 77.70 | 71.44 | 47.25 | 36.92 | 73.19 | 74.70 | 96.55 | 61.46 | 67.40 |
| | CtM (ours) | 77.22 | 69.89 | 51.42 | 36.73 | 74.24 | 75.43 | 96.25 | 73.53 | 69.34 |

### D.7. Scalability to Larger LoRA Collections

To evaluate generalization when merging a larger number of adapters, we extend the setup in Table 1 by adding eight additional datasets: Caltech101 (Fei-Fei et al., 2004), CUB-200-2011 (Wah et al., 2011), Flowers102 (Nilsback & Zisserman, 2008), OfficeHome (Venkateswara et al., 2017), Oxford-IIIT Pet (Parkhi et al., 2012), Food101 (Bossard et al., 2014), FER2013 (Goodfellow et al., 2013), and FashionMNIST (Xiao et al., 2017). We trained one LoRA adapter per dataset. All adapters use input rank $r_{in}=16$ and are fine-tuned with learning rate $3\times10^{-4}$, weight decay 0.1, batch size 32, no label smoothing, a linear warm-up of 500 steps, and at most 50,000 optimization steps with early stopping; all other training and evaluation details follow Appendix C. The target rank $r$ is also set to 16, consistent with the main experiment.

In Table 12, we report the detailed results on all 16 datasets. We also include the normalized accuracy of the base model to better illustrate the difficulty of this larger-scale setting and the gains achieved by different merging methods.

From the results, we find that the larger-scale setting is substantially more challenging: task interference becomes stronger and the gains of standard single-adapter merging methods over the base model are modest. The base CLIP model already attains an average normalized accuracy of 71.80, while the strongest competing single-LoRA-output baseline (Iso-C in CoreSpace) reaches only about 72.45. In contrast, CtM combined with DARE-TIES achieves 73.08 average normalized accuracy, giving roughly +1.28 points over the base model and about +0.68 points over the best baseline. At the same time, the full-parameter Iso-C baseline in CoreSpace attains 77.96 before truncation, which we view as an upper bound that highlights remaining headroom when many tasks are merged. Overall, CtM scales gracefully to $T=16$ adapters and continues to outperform MtC-style and low-rank-aware baselines in this more challenging regime.

*Table 12.* Normalized accuracies on 16 vision tasks with ViT-B/32.

| Method | Space | Cars | DTD | Euro SAT | GTSRB | MNIST | RE SISC | SUN 397 | SVHN | Caltech 101 | CUB 200 | Flowers 102 | Office Home | Oxford-IIIT Pet | Food 101 | FER 2013 | Fashion MNIST | Avg | $Avg_{full}$ |
|---|---|---|---|---|---|---|---|---|---|---|---|---|---|---|---|---|---|---|---|
| *LoRA's absolute accuracy* | | *74.00* | *58.30* | *99.00* | *92.70* | *99.30* | *88.40* | *64.50* | *96.20* | *95.43* | *62.53* | *85.40* | *88.61* | *90.46* | *84.87* | *69.55* | *94.00* | *–* | *–* |
| Base Model | | 80.33 | 75.55 | 46.32 | 34.75 | 48.35 | 68.25 | 98.03 | 32.63 | 89.11 | 81.86 | 75.28 | 98.71 | 93.74 | 98.23 | 57.38 | 70.21 | 71.80 | – |
| LoRA-LEGO | – | 79.97 | 73.63 | 45.30 | 38.29 | 48.97 | 68.68 | 97.76 | 34.10 | 89.67 | 82.79 | 75.91 | 98.79 | 93.03 | 98.33 | 57.55 | 72.78 | 72.22 | – |
| RobustMerge | – | 80.12 | 74.72 | 46.84 | 37.88 | 49.86 | 68.70 | 98.03 | 33.91 | 89.86 | 82.79 | 75.70 | 98.95 | 93.37 | 98.44 | 56.52 | 71.80 | 72.34 | – |
| Iso-C | Full | 80.27 | 74.18 | 43.58 | 37.77 | 50.76 | 68.68 | 97.91 | 34.91 | 89.45 | 81.89 | 75.89 | 99.03 | 93.74 | 98.55 | 58.05 | 72.61 | 72.44 | 77.92 |
| | KnOTS | 79.87 | 74.81 | 44.41 | 37.11 | 48.64 | 68.63 | 97.91 | 32.75 | 89.79 | 82.62 | 75.99 | 98.87 | 93.48 | 98.42 | 57.25 | 72.39 | 72.06 | 72.83 |
| | CoreSpace | 80.31 | 74.27 | 43.55 | 37.76 | 50.91 | 68.72 | 97.90 | 34.94 | 89.48 | 83.52 | 75.89 | 98.95 | 93.78 | 98.53 | 58.10 | 72.65 | 72.45 | 77.96 |
| | CtM (ours) | 79.72 | 76.18 | 46.32 | 44.73 | 50.62 | 72.65 | 98.21 | 38.16 | 89.67 | 82.20 | 74.80 | 98.71 | 93.10 | 98.17 | 54.62 | 69.93 | 72.99 | – |
| TIES | Full | 80.27 | 75.36 | 46.13 | 34.92 | 48.34 | 68.30 | 97.90 | 32.73 | 89.22 | 81.89 | 75.43 | 98.63 | 93.48 | 98.24 | 57.50 | 70.81 | 71.82 | 72.08 |
| | KnOTS | 80.39 | 75.09 | 45.94 | 35.42 | 48.65 | 68.52 | 97.86 | 32.94 | 89.26 | 82.07 | 75.47 | 98.71 | 93.40 | 98.22 | 57.33 | 71.36 | 71.91 | 72.27 |
| | CoreSpace | 80.03 | 75.27 | 46.02 | 35.64 | 49.26 | 68.65 | 97.92 | 33.17 | 89.37 | 83.10 | 75.78 | 99.03 | 93.48 | 98.27 | 57.98 | 72.62 | 72.22 | 74.16 |
| | CtM (ours) | 79.74 | 74.27 | 48.56 | 37.20 | 53.69 | 68.97 | 97.83 | 38.29 | 89.67 | 82.14 | 74.04 | 98.79 | 93.97 | 97.67 | 60.66 | 73.27 | 73.05 | – |
| DARE-TIES | Full | 80.29 | 75.27 | 46.20 | 35.09 | 48.36 | 68.30 | 97.91 | 32.81 | 89.18 | 81.93 | 75.40 | 98.63 | 93.52 | 98.23 | 57.60 | 70.70 | 71.84 | 72.15 |
| | KnOTS | 80.46 | 75.27 | 46.13 | 35.11 | 48.48 | 68.38 | 97.90 | 32.77 | 89.11 | 81.86 | 75.47 | 98.63 | 93.52 | 98.26 | 57.25 | 70.78 | 71.84 | 72.34 |
| | CoreSpace | 80.03 | 75.27 | 45.90 | 37.26 | 49.13 | 68.93 | 97.47 | 33.44 | 89.45 | 83.45 | 75.87 | 99.35 | 93.59 | 98.24 | 56.42 | 72.90 | 72.29 | 74.16 |
| | CtM (ours) | 79.80 | 73.63 | 48.63 | 37.23 | 53.95 | 68.97 | 97.91 | 38.72 | 89.67 | 82.07 | 73.87 | 98.79 | 94.01 | 97.52 | 60.98 | 73.50 | 73.08 | – |

### D.8. Experiments on NLG Tasks

We conduct additional NLG experiments on LLaMA2-7B. Following the evaluation benchmark released with Lo-raRetriever (Zhao et al., 2024), we consider five tasks in our evaluation: ARC-Challenge (Clark et al., 2018), WSC (Levesque, 2011), Glue-QQP (Wang et al., 2019), E2E-NLG (Novikova et al., 2017), WebNLG (Gardent et al., 2017). We adopt the instruction-style input/target formatting and the task-specific evaluation protocols provided by LoraRetriever. For each task, we use the corresponding fine-tuned LoRA adapters for LLaMA2-7B released in LoraRetriever. The rank $r_{in}$ is 8, and the target rank $r$ is also set to 8. Since TIES is a competitive and representative approach in our main experiments, we instantiate our CtM framework with **TIES** as the underlying merge operator, and compare our approach against TIES applied in three different parameter spaces: (i) Full space, (ii) KnOTS space, and (iii) CoreSpace. Additionally, we compare against representative low-rank-aware methods, including LoRA-LEGO, RobustMerge, and Iso-CTS.

As shown in Table 13, CtM consistently yields better average normalized performance across single-LoRA-output baselines, suggesting that CtM is also applicable to generative backbones.

*Table 13.* Normalized performance on 5 NLG tasks with LLaMA2-7B.

| Method | Space | ARC-Challenge | WSC | Glue-QQP | E2E-NLG | WebNLG | Avg |
|---|---|---|---|---|---|---|---|
| *Metric* | | *EM* | *EM* | *EM* | *ROUGE 1* | *ROUGE 1* | |
| *LoRA's absolute performance* | | *46.00* | *50.00* | *74.00* | *66.20* | *70.90* | *–* |
| LoRA-LEGO | – | 108.70 | 100.00 | 91.89 | 88.67 | 70.24 | 91.90 |
| RobustMerge | – | 56.52 | 92.00 | 24.32 | 76.44 | 56.56 | 61.17 |
| Iso-CTS | – | 104.35 | 116.00 | 75.68 | 62.39 | 73.34 | 86.35 |
| | Full | 113.04 | 96.00 | 67.57 | 96.83 | 79.83 | 90.65 |
| TIES | KnOTS | 117.39 | 108.00 | 86.49 | 92.75 | 73.76 | 95.68 |
| | CoreSpace | 113.04 | 100.00 | 83.78 | 99.70 | 79.13 | 95.13 |
| | CtM (ours) | 113.04 | 104.00 | 89.19 | 97.12 | 78.98 | 96.47 |

## D.9. Experiments on Multimodal Tasks

We additionally evaluate CtM on the multimodal benchmark MM-MergeBench (Zeng et al., 2025). Following the benchmark setting, we use LLaVA-1.5-7B (Liu et al., 2023) as the base model and consider eight multimodal generative tasks. These tasks cover diverse multimodal abilities, including question answering, grounding, classification, and captioning.

Following MM-MergeBench, we use task-specific rank-16 LoRAs on LLaVA-1.5-7B, with LoRA modules injected into all linear layers. We merge the eight task-specific LoRAs into a single rank-16 LoRA and compare CtM against the corresponding MtC baselines under three base merge rules: Iso-C, TIES, and DARE-TIES. Following the original benchmark, we report raw accuracies rather than normalized accuracies. For hyperparameter tuning, we randomly select 5% of the benchmark test set to serve as the validation set.

Table 14 summarizes the results. CtM consistently improves over the compared baselines in this multimodal generative setting, showing that the advantages of CtM extend beyond the classification-oriented settings.

*Table 14.* Average performance on the eight seen tasks of MM-MergeBench using LLaVA-1.5-7B. We merge eight rank-16 task-specific LoRAs into a single rank-16 LoRA. CtM consistently outperforms the compared baselines across all three base merge rules.

| Method | Full | KnOTS | CoreSpace | CtM(ours) |
|---|---|---|---|---|
| Iso-C | 56.21 | 56.30 | 56.19 | **56.65** |
| TIES | 54.57 | 56.06 | 56.09 | **56.46** |
| DARE-TIES | 54.71 | 56.16 | 55.42 | **56.54** |

