# OpenReview forum: "Compress then Merge: From Multiple LoRAs into One Low-Rank Adapter"
_ICML.cc/2026/Conference — ICML 2026 regular_

### Official Review · Reviewer_QVVC · 2026-03-07

**Soundness:** 2
**Presentation:** 3
**Significance:** 3
**Originality:** 2
**Overall Recommendation:** 4
**Confidence:** 4

**Summary:**

This paper studies the problem of merging multiple homogeneous LoRA adapters into a single rank-constrained LoRA without using auxiliary data. The authors argue that the standard Merge-then-Compress (MtC) pipeline is structurally mismatched to this goal because it first merges in the full parameter space and only afterward imposes the target rank via truncated SVD, which is optimal for Frobenius reconstruction error but not necessarily for average cross-task utility. In response, the paper proposes Compress-then-Merge (CtM), which first learns shared left and right rank-r subspaces from the input LoRA updates through a reconstruction objective with task reweighting, projects each adapter into an r x r coordinate system, applies a standard merge rule in that reduced space, and lifts the merged coordinates back to the original parameter space. The method also introduces a core-space formulation intended to preserve the optimization objective while reducing computational cost.

The empirical evaluation uses CLIP ViT-B/32 on eight vision tasks and LLaMA3-8B on six NLI tasks, with comparisons against both MtC-style baselines and methods that directly output low-rank updates. The reported results show that CtM consistently improves over single-LoRA-output baselines, often reduces the truncation-related degradation seen in MtC pipelines, and in some settings is competitive with or better than corresponding full-parameter merging baselines before truncation. The paper also includes analyses of target rank, the reweighting hyperparameter beta, and the core-space acceleration, plus appendicial extensions to heterogeneous input ranks, more adapters, and an additional NLG setting with LLaMA2-7B.

**Compliance With Llm Reviewing Policy:**

Affirmed.

**Ethical Review Concerns:**

None. I did not flag this paper for ethics review.

**Final Justification:**

Overall Recommendation: 4

After two rounds of rebuttal, I am maintaining my score of 4.

The authors provided a thorough response with substantial additional experiments. My main concerns have been addressed to a satisfactory degree.

The functional projection test is the most valuable addition. The extension to NLI settings confirms the CLIP result: CtM yields stronger worst-task retention than MtC (91.13 vs 66.86 on the worst-retained NLI task), which is a meaningful and more direct form of evidence for the utility-preservation claim. I do want to note, however, that the NLI per-task breakdown requires careful interpretation: MtC is actually higher on 5 of 6 individual tasks, and the mean is driven by MtC's sharp collapse on SICK. The clearest and most accurate takeaway is that CtM is more robust to worst-case task collapse, not uniformly better. I would encourage the authors to phrase this precisely in the revision rather than presenting it as general superiority on retention.

The ablations on the surrogate objective and beta-reweighting are also convincing. The concatenate-SVD and beta=0 experiments together make a credible case that the paradigm-level decision to enforce the rank constraint before merging is the primary source of CtM's gains, with the surrogate objective and reweighting providing incremental additional benefit.

The LLaVA results are now directly shown, but the gains in the multimodal generative setting are modest (e.g., 56.21 to 56.65 under Iso-C), which suggests the advantage may be smaller in generative settings than in the classification settings emphasized in the main paper. This is worth noting in the paper.

The authors' clarification on base-checkpoint heterogeneity is reasonable -- cross-checkpoint alignment is a distinct problem that falls outside the current formulation.

The paper's contribution is strongest and most clearly supported as the best single-LoRA-output method under a strict rank constraint in homogeneous adapter settings. The authors have agreed to revise the framing relative to full-parameter merging and to present coverage as a diagnostic proxy rather than direct evidence of utility preservation. I am satisfied that these revisions will be made.

**Key Questions For Authors:**

1. How much of CtM's gain comes from the change in merge order itself versus the specific surrogate objective and beta-based reweighting used to learn the shared subspaces? A clear answer here would help determine whether the core contribution is the CtM paradigm broadly or a narrower instantiation whose performance depends substantially on a particular weighting heuristic.

2. Can the authors provide stronger evidence that the learned shared subspaces preserve task utility rather than only improving reconstruction balance or energy coverage? This distinction matters for interpreting how strongly the explanatory claims about "utility-preserving subspaces" should be read versus what the experiments directly demonstrate.

3. The paper emphasizes competitiveness with full-parameter merging methods, but on vision the best Avgfull numbers remain materially higher—how do the authors want this comparison to be interpreted, and what is the fairest headline claim? The answer would affect how readers should position CtM: as a practical low-rank compromise or as a broader replacement for dense merging.

4. How robust is the method outside the homogeneous setting assumed in the main formulation, especially when adapters differ in modules, training recipes, or base checkpoints rather than only rank or number? Given how restrictive the stated setup is, this would clarify the true scope of applicability and whether the conclusions should be read as general LoRA-merging guidance or as guidance for a narrower regime.

**Limitations:**

yes

**Strengths And Weaknesses:**

Strengths

- The paper addresses a practically relevant problem: consolidating multiple LoRA adapters into a single low-rank adapter rather than a dense merged update. That focus is well motivated because the target setting explicitly aims to preserve LoRA's deployment and reuse advantages under a strict rank budget.

- The main methodological idea is conceptually clear and reasonably original at the level of framing: instead of treating the rank constraint as a post-processing step, CtM makes subspace selection the central object and merges only after projecting into shared rank-r subspaces. Even if the components used are classical reconstruction tools, the reversal of the merge/compress order is a meaningful design perspective for this problem.

- The paper gives a technically coherent formulation of CtM. The reconstruction objective, the role of the reweighted surrogate targets, the use of Tucker-style factorization, and the core-space reformulation are presented as a connected pipeline rather than a collection of ad hoc steps. Theorems 4.1-4.3 also clarify what is preserved exactly and what is merely an optimization reformulation.

- The empirical results support a narrower but useful claim: CtM is consistently stronger than the compared methods that must output a single low-rank LoRA. This pattern appears on both the vision and NLI benchmarks, and the paper does make a useful distinction between post-compression performance and pre-compression full-update performance through Avg versus Avgfull.

- The analysis section is more informative than a pure benchmark table. The paper attempts to explain where gains may come from by examining truncation gaps, subspace coverage balance, sensitivity to beta, rank scaling, and core-space efficiency, which makes the work more than just a leaderboard comparison.


Weaknesses

- The central diagnosis of MtC is plausible and empirically motivated, but some explanatory statements are stronger than what the paper directly establishes. In particular, the discussion of "misallocated rank budget," "task-relevant information," and filtering of task-specific noise is supported mainly by performance trends and energy-retention style analyses, not by direct causal evidence showing that the learned shared subspaces isolate utility-bearing directions better in a mechanistic sense. This matters because several claims are framed as explanations rather than observations.

- The empirical support is meaningful but still somewhat narrow relative to the breadth of the framing. The main experiments use one vision backbone with eight datasets and one language backbone with six NLI datasets, and the additional robustness evidence is described only briefly in the main paper. That supports the claim that CtM works in these settings, but it is not yet enough to establish broad superiority across LoRA merging scenarios, architectures, task families, or merge rules.

- The comparison to full-parameter merging methods needs careful calibration. The paper appropriately reports Avgfull for MtC baselines, but some of the narrative risks overstating competitiveness because the strongest full-parameter results still remain clearly better in at least some cases, especially on vision, where CtM narrows but does not close the gap to the best dense merge before truncation. This matters for contribution assessment because "competitive with full-weight merging" is a broader claim than "best among single-LoRA-output methods."

- The role of the reweighting scheme is sensible, but its justification remains heuristic. The paper argues that beta balances equalized participation and scale preservation, yet there is limited evidence that this particular interpolation is the right formulation rather than one reasonable choice among many. Because the method relies on this surrogate objective to learn shared subspaces, stronger ablations on alternative normalization or weighting strategies would help determine how essential this design really is.

- Some concepts that should remain distinct are occasionally blurred. The paper moves between reconstruction quality, subspace coverage, average normalized accuracy, balanced task retention, and utility-preserving subspaces. These quantities are related, but not equivalent. As written, the narrative sometimes invites the reader to treat higher retained energy or lower variance in coverage as stronger evidence of preserving semantic task utility than the experiments strictly show.

---

> ### Author Rebuttal · Authors · 2026-03-31
>
> Thanks for your detailed and constructive comments. We address your concerns point by point as follows:
>
> > **Q1/W4**: How much of CtM's gain comes from the change in merge order itself versus the specific surrogate objective and beta-based reweighting?
>
> **A1**: Thanks for your question. Our additional ablations suggest two complementary points:
>
> - The surrogate objective: We replace the learned subspace with a simple concatenate-SVD subspace, and CtM still remains clearly above the strongest MtC baseline (67.42 vs 65.33 for TIES, 69.04 vs 65.91 for DARE-TIES) (Details in response to Reviewer sFDV, A2). We also replace Frobenius normalization with nuclear/spectral norm, an find it continues to perform well (Details in Reviewer KBpt, A1).
>
> - The reweighting technique:  In Fig. 5 of our original paper, we ablate the role of β in CtM and the results show that, even set β to default value 0, CtM outperforms the best MtC baseline. Moreover, as shown in Appendix D.3, even we employ the same β-reweighting to MtC, the best results only improved only modestly (65.33->66.80), and it still remained substantially below CtM (69.75).
>
> Taken together, these results suggest that the primary source of CtM’s gains is the paradigm-level decision to enforce the rank constraint before merging, while the surrogate objective and β-based reweighting provide additional benefits.
>
> > **Q2/W1**: Providing stronger mechanistic evidence for CtM
>
> **A2**: Thank you for the suggestion. To provide stronger evidence that the shared subspace preserves task utility, we extend the analysis in Section 5.3 with a functional evaluation.  For MtC or CtM, we extract the shared rank-r subspace from the merged LoRA, project each task-specific LoRA onto it, and evaluate the projected LoRA on its own task. We measure **accuracy retention** as the ratio between the accuracy of the projected LoRA and that of the original LoRA.
>
> As shown below, CtM consistently outperforms MtC: **mean accuracy retention** improves from **86.54** to **96.18**,  and **worst-task retention** rises from **47.45** to **93.32**. This provides more direct functional evidence that CtM preserves task utility more faithfully. We will add these results to the revision.
>
> |Method|Cars|DTD|EuroSAT|GTSRB|MNIST|RESISC|SUN397|SVHN|
> |-|-|-|-|-|-|-|-|-|
> |MtC|96.87|79.76|98.96|47.45|99.80|76.89|93.47|99.09|
> |CtM|93.32|93.92|99.41|94.03|99.70|95.49|95.28|98.27|
>
> > **Q3/W3**: Refining the scope and positioning of our claims
>
> **A3**:  Thank you. We agree that our current wording can overstate this comparison, and we will recalibrate it. Our goal is **not** to position CtM as a general replacement for dense/full-parameter merging. Rather, full-parameter merging serves mainly as a reference point for illustrating the truncation gap in MtC pipelines, whereas our target setting is stricter: methods that must output a single fixed-rank LoRA. Under this interpretation, the fairest headline claim is: **CtM is consistently strongest among single-LoRA-output methods, while substantially narrowing the gap to full-weight merging under a strict rank constraint.** We will revise the abstract/introduction/conclusion accordingly and replace broader wording such as “competitive with full-weight merging” with this narrower and more accurate positioning.
>
> > **Q4/W2**: Extending validation to broader experimental settings outside the homogeneous setting, tasks, architectures.
>
> **A4**:  Thanks for the question. During rebuttal, we add the following experiments:
>
> - Experiments beyond homogeneous setting: We use six publicly released LoRAs  which finetune a RoBERTa-base on six GLUE datasets. These LoRAs are trained with different training recipes (batch size, learning rate, epoch), and have different ranks (8/16/32). We merge them into a rank-8 LoRA and report the average normalized accuracy as follows. The results show that CtM remains effective and outperforms MtC under such challenging heterogeneous ranks and training recipes.
>
> ||Full|KnOTS|CoreSpace|CtM|
> |-|-|-|-|-|
> |Iso-C|80.59|80.29|80.58|87.50|
> |TIES|71.65|73.71|71.27|81.23|
> |DARE-TIES|71.77|74.01|75.44|81.86|
>
> - Experiments on different models and tasks: We also add results on LLaVA with 8 multimodal generative tasks and on T5-base with 4 GLUE tasks (see Reviewer sFDV, A1). While the gains vary across settings, CtM consistently outperforms the compared MtC baselines in all tested scenarios.
>
> >**W5**: Conceptual precision and presentation
>
> **A5**: Thank you for pointing this out. We agree that coverage/retained energy and downstream utility are not equivalent. In our paper, coverage is intended as a diagnostic proxy for how evenly the learned subspace captures the original updates. To make this connection more direct, we additionally evaluate a functional projection test in the above response A2. We will add this result in the revision and revise the wording to present coverage/energy as supporting diagnostics, not direct proof of utility preservation.

---

> > ### Author Rebuttal · Reviewer_QVVC · 2026-04-01
> >
> > Thanks for the thorough rebuttal and for adding new experiments.
> >
> > The functional projection test is the most useful addition in my view. The jump in worst-task accuracy retention from 47.45 to 93.32 provides much more direct evidence, at least on the vision side, that the learned subspaces preserve task-relevant information. I still think this claim would be stronger if the same analysis were shown for the NLI or NLG settings, since the current evidence is limited to CLIP.
> >
> > The ablations are also helpful. Concatenate-SVD subspaces remain stronger than MtC, beta=0 already outperforms the best MtC baseline, and adding beta-reweighting to MtC brings only limited improvement. This makes the case that the gain is mainly from the paradigm rather than a particular weighting trick.
> >
> > The new heterogeneous RoBERTa experiment is a useful step, but it does not fully address my original question, which also asked about heterogeneity in adapter modules or base checkpoints, not only training recipes and ranks. The LLaVA and T5 results were referenced but not shown directly here, so I cannot fully assess that part of the scope expansion from the rebuttal alone.
> >
> > I appreciate the willingness to narrow the positioning relative to full-parameter merging and present coverage as a diagnostic proxy. Please make sure these changes are reflected in the final version.
> >
> > Overall, my main concerns about mechanistic support and the paradigm-level contribution are largely addressed. The remaining gaps are about completeness of validation rather than fundamental issues. I will keep my score at 4 for now.

---

> > > ### Author Response · Authors · 2026-04-02
> > >
> > > Thank you for the helpful and timely follow-up. We are glad that the additional functional projection analysis and ablations helped clarify both the mechanistic support and the paradigm-level contribution. We will make sure that the narrowed positioning relative to full-parameter merging, the presentation of coverage as a diagnostic proxy, and the additional analyses are reflected clearly in the final version.
> > >
> > > We address the remaining points below.
> > >
> > > > Functional projection analysis beyond CLIP
> > >
> > > Following your suggestion, we extend the functional projection test to the NLI setting. As in the vision setting, for MtC and CtM we extract the low-rank subspace from the merged LoRA, project each task-specific LoRA onto it, and evaluate the projected LoRA on its own task.
> > >
> > > ||SNLI|MNLI|SICK|QNLI|RTE|SCITAIL|Mean|Min|
> > > |--|---|---|---|---|---|----|---|---|
> > > |MtC|99.84|98.77|66.86|98.93|95.16|99.90|93.24|66.86|
> > > |CtM|99.04|91.74|98.60|98.26|91.13|99.81|96.43|91.13|
> > >
> > > Consistent with the results on CLIP, CtM again yields stronger functional retention than MtC (96.43 vs. 93.24), especially on the worst-retained task  (91.13 vs. 66.86). This provides additional evidence beyond CLIP that CtM yields more balanced functional retention, with notably stronger worst-task preservation than MtC.
> > >
> > > > Directly showing the LLaVA and T5 results
> > >
> > > Thank you for pointing this out. These results were referenced in our first-round rebuttal but not shown directly there due to the character limit. We list them explicitly here for completeness.
> > >
> > > **LLaVA-1.5-7B (MM-MergeBench).** We evaluate CtM on MM-MergeBench [RobustMerge, 2025], an 8-task multimodal generative benchmark, using LLaVA-1.5-7B with LoRAs injected into all linear layers. We merge 8 rank-16 LoRAs into one rank-16 LoRA and report average performance:
> > >
> > > ||MtC (Full Space)|MtC (KnOTS Space)|MtC (CoreSpace)|CtM|
> > > |-----|--------|---------|--------|---|
> > > |Iso-C|56.21|56.30|56.19|56.65|
> > > |TIES|54.57|56.06|56.09|56.46|
> > > |DARE-TIES|54.71|56.16|55.42|56.54|
> > >
> > > **T5-base (high-conflict GLUE subset).** Prior work [FusionBench, 2024] identifies strong conflict among several GLUE tasks on T5-base (CoLA, MNLI, MRPC, STSB). We therefore evaluate CtM on the four released LoRAs.
> > >
> > > ||MtC (Full Space)|MtC (KnOTS Space)|MtC (CoreSpace)|CtM|
> > > |-----|--------|---------|--------|---|
> > > |Iso-C|64.23|60.11|64.24|64.57|
> > > |TIES|60.43|68.93|60.33|72.61|
> > > |DARE-TIES|60.59|70.00|59.87|72.28|
> > >
> > > Taken together, these results extend the evidence beyond the original CLIP and NLI setups: across a multimodal generative benchmark and a different text architecture with a high-conflict task group, CtM continues to outperform the corresponding MtC baselines.
> > >
> > > > Heterogeneity in adapter modules and base checkpoints
> > >
> > > Following your suggestion, we additionally test **within-model adapter-module heterogeneity**. Based on our original ViT setting, we finetune four LoRAs with LoRA modules injected into **``{q, v, mlp.fc1, mlp.fc2}``** on the corresponding datasets (EuroSAT, MNIST, RESISC and SVHN), while keeping the other four LoRAs on the standard **``{q, k, v, o}``** modules. For layers that are not present in all adapters (**``{mlp.fc1, mlp.fc2, k, o}``**), we merge only the updates from the adapters that contain that layer. We then merge all eight adapters and obtain:
> > >
> > >
> > >
> > > ||MtC (Full Space)|MtC (KnOTS Space)|MtC (CoreSpace)|CtM|
> > > |-----|--------|---------|--------|---|
> > > |Iso-C|69.41|63.32|69.42|69.58|
> > > |TIES|64.34|64.76|66.95|69.30|
> > > |DARE-TIES|64.33|64.90|67.40|69.34|
> > >
> > > These results suggest that CtM remains effective under within-model target-module heterogeneity, extending the validation beyond the strictly homogeneous module setting.
> > >
> > > For **base-checkpoint heterogeneity**, we would like to clarify that this lies outside our current formulation. When base checkpoints differ, LoRA updates are no longer expressed relative to a common reference parameterization. The problem therefore is no longer purely LoRA merging, but also involves cross-checkpoint weight alignment/merging. These two components, base-model merging/alignment and LoRA merging, are conceptually distinct, and our paper focuses on the latter under a shared base checkpoint. This is not a simple robustness extension of the current setting, but requires solving a different alignment problem at the base-model level.
> > >
> > > In summary, we (i) extend the validation to NLI functional retention, (ii) directly present the previously referenced LLaVA and T5 results, (iii) provide additional evidence for target-module heterogeneity within a shared base model, and (iv) clarify that broader cross-checkpoint heterogeneity falls outside the current formulation and remains an important direction for future work. We hope these additions help address the remaining concerns about validation completeness and further clarify both the scope and the empirical support of our work.

---

### Official Review · Reviewer_KBpt · 2026-03-09

**Soundness:** 2
**Presentation:** 3
**Significance:** 2
**Originality:** 2
**Overall Recommendation:** 4
**Confidence:** 4

**Summary:**

This paper explores the practical problem of consolidating multiple task-specific LoRA adapters into a single rank‑$r$ LoRA adapter without using data, motivated by deployment and reuse constraints. The authors argue that the common Merge‑then‑Compress (MtC) pipeline, which consists of merging in full space then compressing with truncated SVD, allocates the limited rank budget poorly (e.g., dominated by large‑norm adapters) and can yield brittle per-task degradation. They propose Compress‑then‑Merge (CtM): first learn shared left/right rank‑$r$ subspaces $U,V$ by solving a multi-matrix reconstruction objective (implemented via Tucker/HOSVD+HOOI) with a reweighting parameter $\beta$; then project each adapter into $r\times r$ coordinates, apply an existing merge rule, and lift back to obtain a rank‑$r$ update by construction. Experiments on CLIP ViT‑B/32 and LLaMA3‑8B show clear improvements of CtM over MtC.

**Compliance With Llm Reviewing Policy:**

Affirmed.

**Final Justification:**

I thank the authors for a clear and constructive rebuttal. The additional experiments, especially the normalization ablations (Frobenius vs. nuclear vs. spectral) and the heterogeneous LoRA setting, strengthen the empirical case that CtM provides consistent gains over MtC across a range of conditions. I also appreciate the clarification that β-reweighting only affects subspace learning and not the merge itself, which helps sharpen the conceptual distinction between CtM and reweighted MtC.

That said, my main concerns remain only partially addressed.

First, while the new ablations suggest that CtM’s improvements are not tied to a specific normalization, they still do not fully disentangle the core claim that “rank constraint before merge” is the key driver. In particular, the comparison space for MtC remains somewhat limited (e.g., no stronger or jointly optimized weighting/normalization strategies beyond β-reweighting), so it is still difficult to attribute gains cleanly to the ordering of operations rather than to the particular subspace construction.

Second, I appreciate the clarification that Theorems 4.1-4.3 are intended to justify the core-space acceleration rather than to provide novel theory. However, this reinforces my original assessment that the theoretical contribution is mainly supportive rather than advancing understanding of the problem itself.

Third, the revised positioning around full-weight merging is helpful and more accurate. However, the fact that CtM can surpass dense merging in some cases still points to an interesting (and potentially important) phenomenon, e.g., interference or implicit regularization, that remains unexplored. While I agree this is not required for the paper to be valid, it does limit the depth of insight provided.

Overall, I view the paper as technically solid, well-motivated, and practically relevant, with a simple and useful method that improves over existing baselines in the constrained setting of fixed-rank LoRA merging. At the same time, the empirical and conceptual evidence is not yet strong enough to fully substantiate the central claims at a deeper level or to elevate the contribution beyond an incremental but useful improvement.

For these reasons, I maintain my score of 4.

**Key Questions For Authors:**

Does CtM still help when the input LoRAs are trained with different optimizers / different ranks / different scaling beyond the narrow appendix?

**Limitations:**

yes

**Strengths And Weaknesses:**

### Strengths
- The paper addresses a real deployment constraint by producing a single fixed-rank LoRA adapter.
- The motivating analysis of MtC and conceptual framing of where MtC can fail are clear and easy to follow. The method proposed to resolve these issues is simple and implementable, and the core space acceleration is a good engeneering contribution.
- The paper persents a broad evaluation across vision + language, plus extra appendices, making for a mostly (see weaknesses why only "mostly") convincing evaluation of the proposed method.
### Weaknesses
- The convex combination of the adapter's Frobenius norm and the averaged Frobenius norm seems ad hoc. Why did the authors decide to normalize by Frobenius norm at all (vs spectral norm, layerwise normalization, per-module normalization)? Additionally, the authors' critique of MtC is “SVD favors large-norm tasks”, but MtC pipelines could incorporate weighting before merge or before truncation. The paper (including the small control study in the appendix) does not convincingly establish that CtM’s gains come from “rank constraint before merge” rather than from “we used a particular normalization”.
- Theorems 4.1-4.3 are essentially straightforward statements about projection onto the span of concatenated LoRA factors and equivalence of optimizing in that subspace. They are correct but not a meaningful theoretical advance.
- The fact that CtM sometimes surpasses full-weight merging in the experiments suggests that either (i) the unconstrained merge is poorly tuned / not comparable, (ii) validation selection differs in a way that favors CtM, or (iii) the “full merge” is not truly an upper bound due to interference. The authors provide an interpretation (“filters noise”), but do not actually test it (e.g., analyze whether the full merge is overfitting).

---

> ### Author Rebuttal · Authors · 2026-03-31
>
> We thank the reviewer for the constructive suggestions. We respond to each point below.
>
> > **W1**: About the choice of Frobenius norm and the gains of CtM
>
> **A1**: We thank the reviewer for this important question. We chose Frobenius normalization as a simple matrix-level way to reduce large-norm dominance when learning the shared subspaces. We agree that this choice is not uniquely mandated, and stronger normalization ablations are helpful. To address this, we additionally evaluated CtM with alternative normalizations. As shown below, CtM remains better than the strongest MtC baseline under nuclear and spectral norms as well, indicating that the improvement is not tied to a single normalization choice.
>
> ||Iso-C|TIES|DARE-TIES|
> |-|-|-|-|
> |Best MtC|65.72|65.33|65.91|
> |CtM (Frobenius Norm)|66.27|69.75|70.68|
> |CtM (Nuclear Norm)|66.22|69.27|69.61|
> |CtM (Spectral Norm)|66.12|67.88|67.52|
>
> We agree that MtC can also incorporate weighting techniques to help merging. That's why we employed the same β-reweighting to MtC in Figure 6. The results show that, the best reweighted MtC baseline improved only modestly (65.33->66.80), and it still remained substantially below CtM (69.75).
>
> We want to clarify an important point in our formulation: β-reweighting affects only the learning of the shared subspaces. The coordinates used for merging are always computed from the original, unscaled LoRA updates. Thus, β influences which directions survive the rank-$r$ bottleneck, rather than acting as a direct task reweighting during merging.
>
> Overall, these results suggest that CtM’s gains mainly come from enforcing the rank constraint before merging, rather than from any particular normalization heuristic alone.
>
> > **W2**: Theorems 4.1-4.3 are correct but not a meaningful theoretical advance.
>
> **A2**: The purpose of 4.1-4.3 is to support the core-space acceleration in Sec. 4.3. Although they do not represent a major theoretical advance, they are critical to the algorithm design:
>
> - Theorem 4.1 shows lossless projection of all LoRA updates into the span of concatenated factors.
> - Theorem 4.2 shows that an optimal solution exists within this core space.
> - Theorem 4.3 establishes equivalence between the reduced and original optimization problems.
>
> Thus, the acceleration reduces computation without changing the optimum. Empirically, Table 5 shows a reduction from 690.83s to 20.77s while preserving the same average normalized accuracy (94.15 vs. 94.15). We will revise the paper to make clearer that these theorems are included to support the acceleration’s correctness, rather than to claim a major theoretical contribution.
>
> > **W3**: The fact that CtM sometimes surpasses full-weight merging in the experiments is not deeply analyzed.
>
> **A3**: Thank you for pointing this out. We also found it interesting that CtM can sometimes outperform the full-weight merging results. Our intention was only to share one possible intuition rather than to claim a validated mechanism, and we agree that the current wording sounds stronger than what our evidence directly supports.
>
> More importantly, our focus is the **single fixed-rank LoRA output** setting. We therefore do **not** intend to claim that CtM generally outperforms dense merging; indeed, on vision, the strongest dense baseline is still clearly better. Our intended claim is narrower: among methods that must output a single rank-constrained LoRA, CtM performs best while substantially narrowing the gap to full-weight merging. We will revise the discussion accordingly to sharpen the positioning to avoid possible misunderstanding.
>
>
> > **Q1**: Does CtM still help when the input LoRAs are trained with different optimizers / different ranks / different scaling beyond the narrow appendix?
>
> **A4**: Thanks for the question. We provide the additional experiments here. We use six public LoRAs  released on Huggingface, which finetune a RoBERTa-base on corresponding six GLUE datasets (MNLI, QNLI, QQP, RTE, SST2, and STSB). These LoRAs are trained with different training recipes (batch size, learning rate and epoch), have different ranks (8/16/32) and scaling factors. We then merge them into a rank-8 LoRA and report the average normalized accuracy as follows. The results show  that CtM remains effective and outperforms MtC under such challenging heterogeneous ranks and training recipes.
>
> ||Full|KnOTS|CoreSpace|CtM|
> |-|-|-|-|-|
> |Iso-C|80.59|80.29|80.58|87.50|
> |TIES|71.65|73.71|71.27|81.23|
> |DARE-TIES|71.77|74.01|75.44|81.86|

---

> > ### Author Rebuttal · Reviewer_KBpt · 2026-04-02
> >
> > I thank the authors for their detailed response. All my concerns have been addressed. I decided to keep my score at 4.

---

> > > ### Author Response · Authors · 2026-04-04
> > >
> > > Thank you for your thoughtful follow-up and positive assessment.
> > >
> > > We are especially encouraged that you recognized the real deployment relevance of our rank-constrained setting, the clarity of the CtM framing relative to MtC, and the value of the core-space acceleration and cross-domain evaluation.
> > >
> > > We are also very glad that our rebuttal has adequately addressed your concerns. The additional evidence further strengthens the practical motivation and methodological clarity you identified in the original submission, while also broadening the paper’s empirical support.
> > >
> > > We will incorporate the key clarifications and added results into the final version so that these contributions are presented even more clearly.

---

### Official Review · Reviewer_i997 · 2026-03-13

**Soundness:** 3
**Presentation:** 3
**Significance:** 3
**Originality:** 2
**Overall Recommendation:** 4
**Confidence:** 4

**Summary:**

The paper tackles the problem of merging multiple task-specific LoRA adapters into a single, fixed‑rank LoRA adapter while preserving efficiency and modular deployment. It highlights a core issue with the standard Merge‑then‑Compress (MtC) strategy: merging adapters first and applying truncated SVD afterward often yields suboptimal results, because post‑hoc compression wastes limited rank capacity on large‑magnitude or noisy directions rather than those important for multi‑task performance. To address this, the authors propose Compress‑then‑Merge (CtM), a new framework that enforces the target rank before merging.

The core methodology involves several key steps:

**Step 1: Shared Subspace Learning (Eq 2):** The method learns optimal r-dimensional left and right subspaces that capture common structures across all input adapters by optimizing a reconstruction objective.

**Step 2: Rescaling Mechanism (Eq 4):** To ensure the learned subspaces reflect broadly useful directions rather than just the most dominant tasks, CtM incorporates a rescaling step to balance the influence of adapters with different norms.

**Step 3: Coordinate Merging (Eq 5-6):** Each adapter (total tasks =T) is projected into the shared subspace, represented as a compact Tr×Tr coordinate matrix M^(t). This is "core-space" acceleration [Panariello et al., 2025], providing substantial speedups in subspace computation without sacrificing performance. Standard merging rules (like TIES or Task Arithmetic) are then applied directly to these coordinates.

**Step 4: Constructive Low-Rank Output (Eq 7):** By lifting the merged coordinates back to the original parameter space [Panariello et al., 2025], the method guarantees a rank-r LoRA update by construction, avoiding brittle post-hoc truncation.


(Panariello et al., 2025): Accurate and efficient low-rank model merging in core space. In NeurIPS, 2025

**Compliance With Llm Reviewing Policy:**

Affirmed.

**Final Justification:**

Based on rebuttal and clarity of scope of work and improvements through additional experiments, I have raised the score!

**Key Questions For Authors:**

Q1: If the authors can provide evidence (or a theoretical bound) showing that the r×r coordinate space can still capture essential features under high divergence (Robustness to Task Orthogonality), it would significantly alleviate concerns regarding the "rigid bottleneck" and broaden the paper’s perceived significance.

Q2: Recent works such as RAVAN (NeurIPS '25) and HiRA (ICLR '25) suggest that effective multi-task consolidation actually requires high-rank updates or Hadamard-product structures to avoid information collapse. Why is a single, fixed-rank LoRA sufficient in authors' view, and did one consider comparing CtM against these high-rank alternatives?

Q3: Ablation of Subspace Learning vs. Base Rules: How much of the final performance is attributed to the optimal subspace selection (U,V) versus the coordinate-level merging?

**Limitations:**

The authors should include a more comprehensive discussion under Impact Statement.

1. The reliance on searching for optimal β (rescaling) and α (scaling) factors, alongside base-rule parameters (top-K, pruning rates), should be framed as a limitation for "out-of-the-box" deployment.

2. As model merging makes it easier to combine many decentralized adapters, it complicates the ability to attribute model behavior back to a specific data source or original adapter creator. This has implications for copyright and model safety auditing.

3. While merging promotes efficiency, it could also be used to combine "malicious" specialized adapters into a single capable but dangerous model.

4. The framework requires solving optimization problem (Problem 11) for U_{core} and V_{core}. For practical deployments, the impact of this computation should be addressed

**Strengths And Weaknesses:**

### STRENGTHS

1. CtM enforces rank constraints before merging to prevent the information loss inherent in post-hoc SVD-based compression. This ensures the rank budget is allocated to the most critical multi-task features rather than being wasted on task-specific noise.

2. Theorem 4.2 proves that optimal subspaces exist within a "core space," enabling mathematically lossless CoreSpace [Panariello et al., 2025], acceleration.
This provides a formal guarantee that the method’s computational efficiency does not come at the cost of reconstruction accuracy.

3. The framework demonstrates consistent state-of-the-art performance across both Vision (ViT-B/32) and Language (LLaMA3-8B) architectures. These cross-domain results prove that CtM is a versatile, architecture-agnostic solution for modern foundation model specialization.

4. Operating in a shared low-rank subspace acts as a regularizer, filtering out the "chatter" that degrades standard merges.
This selective focus allows the merged low-rank adapter to occasionally surpass the performance of even full-parameter merging benchmarks.

---

### WEAKNESSES

A. The framework’s efficacy hinges on the assumption that distinct task-specific adapters concentrate their changes within a shared, learnable low-dimensional subspace. If the task updates are highly orthogonal or lack structural overlap, the rigid r-dimensional bottleneck could lead to significant and irretrievable information loss. There have been other recent works [RAVAN, NeurIPS'25 ] [HiRA, ICLR'25] [ABBA, ICML'25] which advocate for global adapters constructed from merging local LoRA require high-rank updates.

 - RAVAN: Multi-Head Low-Rank Adaptation for Federated Fine-Tuning, NeurIPS'25

- HiRA: Parameter-efficient hadamard high-rank adaptation for large language models, ICLR'25

- ABBA: Highly Expressive Hadamard Product Adaptation for Large Language Models, ICML'25

B. While CtM mitigates the issues of standard Merge-then-Compress pipelines, a performance gap between pre- and post-merging versions still persists in several benchmarks, especially in vision tasks. This suggests that no training-free method can fully recover the performance of high-rank merged updates within a very small rank budget.

C. CtM acts as a "wrapper" that prepares data for merging, but it still relies on existing, MERGE heuristics for the final step. Hence, CtM is limited by those rules' ability to resolve misaligned conflicts across task adapters.

---

> ### Author Rebuttal · Authors · 2026-03-31
>
> Thank you for your constructive comments. We address your concerns point by point:
>
> > **W2/Q2**: Why is a single, fixed-rank LoRA sufficient in authors’ view, and did one consider comparing CtM against these high-rank alternatives?
>
> **A1**: We do not assume that a single, fixed-rank LoRA is always sufficient for merging. In fact, the gap to unconstrained full-weight merging in several settings suggests that this low-rank bottleneck can incur information loss, which is also consistent with the literature you pointed out.
>
> But considering the rank budget is highly meaningful in resource-constrained deployment. In many practical scenarios, LoRA is adopted precisely because of storage, memory, and serving cost. Moreover, LoRA is a widely adopted interface in existing PEFT and serving ecosystems (e.g., HuggingFace PEFT and vLLM), making a single merged LoRA especially convenient for reuse and deployment.
>
> Recent high-rank or more flexible PEFT alternatives are valuable related directions, but they operate under substantially different output budgets and deployment assumptions. For the same reason, our main experiments focus on methods that must produce a single rank-r LoRA, while full-weight merging is included only as an unconstrained reference to illustrate the truncation gap.
>
> Accordingly, we will revise the paper to **make our claim clearer**: among methods that output a single rank-constrained LoRA, CtM consistently performs best, while substantially narrowing the gap to full-weight merging. We will also discuss the works you provide to expand the discussion of recent high-rank alternatives.
>
> > **W1/Q1**: If the task updates are highly orthogonal or lack structural overlap, the rigid r-dimensional bottleneck could lead to significant and irretrievable information loss.
>
> **A2**: We agree that under an extreme orthogonality scenario, any method constrained to output a single rank-constrained LoRA must incur irreversible information loss. CtM does not remove this bottleneck. Our point is instead comparative rather than absolute: once a strict rank budget is imposed, the key question becomes how to allocate that limited subspace across tasks.
>
> This is precisely where CtM differs from MtC. In MtC, the final rank-r subspace is chosen only after task updates have already been merged and entangled, which can bias the subspace toward some tasks while discarding others. CtM instead makes the bottleneck explicit before merging and optimizes the shared subspace itself. As a result, CtM retains substantially higher per-task utility after projecting each original LoRA onto the learned subspace.
>
> Our empirical results further validate the effectiveness of CtM. Among single-LoRA-output methods, CtM consistently achieves the best performance across various settings. Furthermore, our experiments with larger target ranks in Fig.4 of our original paper demonstrate that while increased capacity naturally improves the performance of all methods, and CtM consistently maintains its lead. In the final version, we will clearly explicitly clarify that our method is designed to optimize performance under a fixed rank constraint; for highly orthogonal tasks, expanding the rank capacity remains a necessary solution.
>
> > **W3/Q3**: Does CtM rely on the base merge rule? How much of the gain comes from subspace selection versus coordinate-level merging?
>
> **A3**: CtM does not define a new merge rule; rather, it is a general paradigm that first learns a shared rank-r subspace and then applies a chosen base merge rule within that space. Therefore, it is complementary to existing merge rules rather than tied to any single one.
>
> Empirically, CtM consistently improves over the corresponding MtC pipeline across representative merge rules as shown in the following table. We also observe larger gains for coordinate-wise rules such as PCB-Merging, TIES, and DARE-TIES, which suggests that developing merge algorithms specifically tailored to CtM may be a promising direction for future work.
>
> ||Task Arithmetic|Iso-C|PCB-Merging|TIES|DARE-TIES|
> |-|-|-|-|-|-|
> |Best MtC|62.63|65.72|63.31|65.33|65.91|
> |CtM|62.82|66.27|66.34|69.75|70.68|
> |Improvement|+0.19|+0.55|+3.03|+4.42|+4.77|
>
> To further isolate the role of the subspace itself, we project each original task LoRA onto the rank-r subspaces induced by MtC and CtM, respectively. CtM preserves both more task energy (87.94 vs. 69.69) and substantially better task utility (96.18 vs. 86.54), showing that its learned shared subspace is already more effective and balanced before attributing gains to any particular coordinate-level merge rule.
>
> Overall, the evidence suggests that the primary structural gain comes from better subspace selection under the rank bottleneck, while stronger merge rules can further exploit that space.
>
> > Limitation discussion.
>
> We thank the reviewer for highlighting these limitations and will strengthen the discussion accordingly, especially regarding extreme task orthogonality.

---

> > ### Author Rebuttal · Reviewer_i997 · 2026-04-04
> >
> > Thanks to the authors for the detailed response and extra results that demonstrate the effectiveness of CtM.
> >
> > Authors must frame their work under the lens of constrained r-rank methods to have better clarity and have separate discussion on higher rank alternatives to enhance their paper!
> >
> > I thereby increase by score to 4.

---

> > > ### Author Response · Authors · 2026-04-04
> > >
> > > Thank you for raising your score and for your constructive feedback. We are very pleased that our additional results and clarifications have fully resolved your concerns.
> > >
> > > Your suggestions have helped us substantially improve the clarity, positioning, and broader perspective of the paper, especially by sharpening the rank-constrained framing and clarifying its relationship to higher-rank alternatives.
> > >
> > > We believe this discussion not only strengthens the presentation, but also better highlights CtM as a promising paradigm for constrained LoRA merging.
> > >
> > > We will make these improvements explicit in the final version so that the paper better reflects both its intended scope and its broader significance.

---

### Official Review · Reviewer_sFDV · 2026-03-13

**Soundness:** 3
**Presentation:** 3
**Significance:** 3
**Originality:** 2
**Overall Recommendation:** 3
**Confidence:** 4

**Summary:**

This paper looks at training-free merging of multiple homogeneous LoRA adapters into a single rank-constrained LoRA. The authors argue that Merge-then-Compress (MtC) is a poor fit for the actual goal because it merges at higher rank and only later forces rank (r) via truncated SVD. They propose Compress-then-Merge (CtM): learn shared rank-(r) left/right subspaces from LoRA weights using a reweighted reconstruction objective, project each adapter into an (r\times r) coordinate space, merge there, and lift back to get a rank-(r) adapter by construction. They also provide a Tucker/HOSVD/HOOI view plus a core-space speedup, and evaluate on CLIP ViT-B/32 vision tasks and LLaMA3-8B NLI, with appendix extensions (heterogeneous ranks, more adapters, NLG).

**Compliance With Llm Reviewing Policy:**

Affirmed.

**Final Justification:**

While the authors' thorough rebuttal effectively addresses my empirical concerns regarding evaluation scope, efficiency, and stability, my fundamental reservation regarding originality remains. The "Compress-then-Merge" (CtM) paradigm, though practically effective, is primarily a logical reordering of operations and a clean engineering integration of existing mathematical tools (Tucker decomposition, HOOI, shared subspaces). It does not introduce a novel algorithmic primitive or a profound conceptual leap. The "paradigm shift" argued by the authors feels incremental rather than groundbreaking. Consequently, despite the solid experimental results, the theoretical novelty does not meet the bar for acceptance, and I am maintaining my score.

**Key Questions For Authors:**

1. **Generality:** Can you show at least one additional backbone/injection setting and one higher-conflict task group to test whether CtM still beats strong MtC baselines?
2. **Ablations:** With merge rule fixed (TIES/DARE-TIES), can you ablate: learned subspace vs simple SVD/PCA subspace; coordinate-space merge vs averaging; with vs without (\beta) reweighting?
3. **Mechanism:** Can you add more direct evidence (e.g., retention–performance correlations, principal-angle stats, and coordinate conflict metrics like sign disagreement/top-k overlap) that actually explain the gains?
4. **Practical efficiency:** Under comparable tuning budgets, how do the full CtM pipeline and MtC merge+truncation compare in wall-clock time and peak memory usage?
5. **Stability & tuning fairness:** Do results hold across ≥3 seeds (mean±std), and are search budgets matched (candidates/step sizes/total trials)?

**Limitations:**

The limitations/societal impact discussion is thin. I’d like a clearer statement of scope (homogeneous adapters; training-free merging), expected failure modes under strong conflict, and practical cautions for safety-sensitive use.

**Strengths And Weaknesses:**

I like the framing: strict low-rank deployment is the real constraint in practice, and the “compress before merge” idea is simple and well-motivated. The method is described clearly enough to implement, and the main results (plus the truncation-gap analysis) support the narrow claim that CtM improves over the compared single-LoRA-output baselines in these settings. Writing is clear; presentation is fine.

My concern is mostly about scope and validation depth. The main evidence is still centered on one vision backbone/setup and one NLI LLM setup; the appendix helps but doesn’t fully convince me CtM is broadly reliable across architectures, injection choices, and harder conflict regimes. The “why it works” story (conflict concentration / noise filtering) is plausible but remains somewhat indirect. On originality, this feels more like a clean integration of known pieces (shared low-rank subspaces + Tucker/HOOI + low-dim merging) than a fundamentally new primitive. I also want clearer reporting on practical overhead and stability (seeds, tuning budget parity) against strong MtC baselines.

---

> ### Author Rebuttal · Authors · 2026-03-31
>
> Thanks for the constructive feedback. We address your concerns point by point:
>
> > Weakness on originality
>
> We agree that CtM does not introduce a new tensor primitive. Its contribution is a bottleneck-first paradigm for fixed-rank many-to-one LoRA merging: under a strict rank budget, the key object is the final shared subspace, since any directions outside it are irreversibly lost. MtC chooses this subspace after merging, when task information is already entangled. CtM makes it explicitly beforehand, allowing us to select the subspace representing cross-task structure.
>
> The techniques in CtM are introduced to make this paradigm work in training-free setting: to obtain the subspace, we design the rescaling-aware surrogate objective; to solve it efficiently, we introduce Tucker and core-space acceleration. Thus, CtM is not just a reordering of known steps, but a change in when and what fixed-rank LoRA merging controls. We will revise the paper to clarify this point.
>
> >Q1: More experiments
>
> **A1**: We add the following experiments:
>
> - A new backbone/injection setting: MM-MergeBench [RobustMerge, 2025] comprises 8 multimodal generative tasks, using LLaVA-1.5-7B with LoRAs on all linear layers. We merge 8 rank-16 LoRAs into one rank-16 LoRA and report avg. performance:
>
> ||Full|KnOTS|CoreSpace|CtM|
> |-|-|-|-|-|
> |Iso-C|56.21|56.30|56.19|56.65|
> |TIES|54.57|56.06|56.09|56.46|
> |DARE-TIES|54.71|56.16|55.42|56.54|
>
> - A high-conflict setting: Prior work [FusionBench, 2024] identified high conflict among several GLUE tasks (CoLA, MNLI, MRPC, STSB) on T5-base. They claim that finetuning on one task can substantially degrade other tasks' performance. We evaluate CtM on the four LoRAs they released:
>
> ||Full|KnOTS|CoreSpace|CtM|
> |-|-|-|-|-|
> |Iso-C|64.23|60.11|64.24|64.57|
> |TIES|60.43|68.93|60.33|72.61|
> |DARE-TIES|60.59|70.00|59.87|72.28|
>
> These results suggest CtM remains effective across new setups and under strong task conflicts.
>
> >Q2: Ablations to isolate CtM’s gains.
>
> **A2**: We provide the following ablations:
>
> - Learned vs. simple SVD subspaces: As shown in the table, using simple SVD subspaces instead of learned ones lowers performance, but still beats the best MtC. We obtain them by vertically/horizontally stacking the normalized adapters and taking the top-r left/right singular vectors. This suggests, simple SVD subspaces already capture much of CtM’s gain, while learned subspaces further improve performance.
>
> |Method|TIES|DARE-TIES|
> |-|-|-|
> |Best MtC|65.33|65.91|
> |CtM+SVD subspace|67.42|69.04|
> |CtM+learned subspace|**69.75**|**70.68**|
> - Role of β: We already ablated β in Fig.5, and show its results here: CtM under the default β=0 setting outperforms the best MtC, while a coarse β sweep improves further. Moreover, transferring the same β-reweighting to MtC yields only limited gains (65.33->66.80), substantially below CtM (69.75) (App. D.3).
>
> ||TIES|DARE-TIES|
> |-|-|-|
> |Best MtC|65.33|65.91|
> |CtM with β=1|65.28|65.48|
> |CtM with β=0|67.49|70.06|
> |CtM with coarse sweep on β|**69.75**|**70.68**|
> - Across base rules, CtM improves over MtC by +0.19/+4.42/+4.77 for averaging/TIES/DARE-TIES, with larger gains for conflict-aware rules.
>
> Overall, the gain mainly comes from enforcing the rank bottleneck before merging, with better subspaces and stronger merge rules providing additional gains.
>
> >Q3: Mechanism analysis.
>
> **A3**: For direct functional evidence, we project each task-specific LoRA onto the final merged rank-r subspace from CtM or MtC, and measure accuracy retention on its own task (projected acc. / original acc.).
>
> CtM consistently outperforms MtC: mean accuracy retention rises from 86.54% to 96.18%, and worst-task retention surges from 47.45% to 93.32%. This supports that CtM preserves task utility more faithfully, not merely reconstruction balance. (Details in A2, Reviewer QVVC)
>
> >Q4: Efficiency comparison.
>
> **A4**: We report end-to-end wall-clock time and peak memory on ViT with TIES under the same linear search protocol. Compared with necessary sweeps over base-rule hyperparameters, CtM adds only one extra hyperparameter, β, searched on a coarse 4-point grid. The added cost is linear rather than exponential, thus CtM remains comparable to MtC in both time and memory as shown.
> |Method|Time(min)|Memory(GB)|
> |-|-|-|
> |Full|35.44|1.05|
> |KnOTS|38.20|1.26|
> |CoreSpace|34.24|1.13|
> |CtM|37.80|1.08|
>
>
> >Q5: Stability and tuning fairness.
>
> **A5**: Our subspace construction (HOSVD+HOOI) is deterministic, so randomness only comes from stochastic base rules. Since TIES and Iso-C are deterministic, we report DARE-TIES over 3 random seeds. CtM remains stable across seeds, with gains well above the variance. We use the same linear search protocol for fairness.
> |Method|ViT|LLaMA|
> |-|-|-|
> |Full|63.41±0.05|90.72±0.01|
> |KnOTS|63.39±0.01|92.70±0.42|
> |CoreSpace|64.66±1.11|92.89±0.01|
> |CtM|70.12±0.56|95.01±0.68|
>
> >Limitations
>
> We thank the reviewer for this suggestion and will strengthen the discussion of limitations accordingly.

---

> > ### Author Rebuttal · Reviewer_sFDV · 2026-04-03
> >
> > Questions are basically answered. But the originality is still not strong enough in the current version. I decide to keep my score at 3.

---

> > > ### Author Response · Authors · 2026-04-03
> > >
> > > We are glad to see your statement that the issues you raised have been fully resolved, though we regret that your overall assessment nevertheless remains unchanged. Due to the response-length constraint, our first-round reply focused primarily on the specific technical concerns you raised. Here, we would like to further clarify the **originality** of our work.
> > >
> > >
> > >
> > > The contribution of this paper does not lie in introducing yet another isolated mathematical tool. Rather, it lies in developing a **new paradigm** at the level of framing for the practical problem of **merging multiple LoRAs into a single fixed-rank LoRA**. Specifically, our main contributions are as follows:
> > >
> > > * **Identify the drawback of existing MtC pipelines**: Under a strict rank constraint, final performance is determined not only by the merged high-rank update, but also by the final rank-$r$ subspace that is retained. In MtC, however, this subspace is not explicitly optimized before merging; instead, it is deferred to post-processing via truncated SVD. As a result, the merge procedure is agnostic to the final rank budget, and the retained subspace is shaped more by the spectral structure of the merged update than by overall cross-task utility. Our analysis of MtC, including the truncation-gap discussion and the functional-retention results, is centered on this structural mismatch.
> > >
> > > * **Propose a bottleneck-first CtM paradigm**: Motivated by this observation, CtM decouples subspace construction from the merge rule itself: it first explicitly learns a shared low-rank subspace, and then performs merging in the corresponding low-dimensional coordinates. In this way, the rank constraint becomes a design object **before** merging, rather than a post hoc consequence after merging. Put differently, our core contribution is to **elevate subspace selection as an explicit optimization target**, rather than a byproduct of MtC. Unlike prior coordinate-system-based merging methods (e.g., KnOTS/CoreSpace), which mainly seek a lossless shared coordinate system, CtM learns an intentionally lossy low-rank shared space under the target rank budget.
> > >
> > > * **Instantiate CtM as a complete training-free pipeline**: The technical design is coherent under this formulation rather than a collection of disconnected components:
> > >   **(i)** We introduce a rescaling-aware shared-subspace objective  **using normalization and $\beta$-weighting techniques**. This effectively deals with an important problem that large-norm task may dominate the subspace, so that the learned subspace better captures structure shared across tasks.
> > >   **(ii)** We formulate the shared-subspace optimization as a Tucker-based solver and combine it with a core-space acceleration, yielding a solution that **is both principled and practical**. This enables stable optimization via HOSVD+HOOI while substantially reducing the cost of decomposition. In this pipeline, we **provide theoretical guarantees** that the core-space reformulation is lossless and preserves optimal solutions of the original shared-subspace problem. As a result, this is not a heuristic compression trick but a practical acceleration, reducing the complexity by a factor of $(n/Tr)^2$, roughly two orders of magnitude in typical settings.
> > >
> > > Both the original experiments and the ones you suggested show that, across multiple backbones, datasets, and LoRA settings, **CtM is consistently more effective than MtC**. The **functional retention** analysis further supports that the shared subspace learned by CtM preserves multi-task utility more evenly than MtC. This perspective also suggests a natural future direction: designing **low-rank-aware merge rules specifically for the CtM coordinate space**, rather than directly reusing rules developed for the full space.
> > >
> > > Therefore, we hope the paper will be understood as follows: for the practical regime of single fixed-rank LoRA consolidation, it proposes a **new bottleneck-first formulation at the level of framing** and instantiates it as a **complete training-free pipeline** that is principled, provable, and computationally efficient. While some components build on established techniques, this instantiation is not a mechanical reuse of prior tools: it introduces a new rescaling-aware objective and provides a principled yet practical solver with theoretical guarantees for lossless acceleration.

---

### Decision · Program_Chairs · 2026-04-30

**Decision:**

Accept (regular)

**Comment:**

The reviewers primary concerns centred around the novelty and strength of the contributions relative to the strong claims (i.e. "paradigm shift") made by the authors, the evidence that the ordering of merging after compression vs. existing compression vs. merging was the primary factor driving improved results over other parts of the proposed methodology, the clear description of the context in which the authors proposed the methodology (i.e. only in the context of requiring a low-rank model rather than claiming to beat dense), a lack of explanation/analysis for some of the more surprising results (e.g. the odd result where dense merging performance was surpassed, and the lack of generative rather than classification results.

Post-rebuttal, most of these concerns appear to have been addressed, except for the strength of contributions which multiple reviewers (even those proposing a weak accept) point to as lacking compared to the claims the authors make. Most of these concerns appear centred around the writing of the paper.

On the writing front, it appears the authors make very strong and potential over-claims about their methodology being a "paradigm shift" throughout the work, and also mislead with claims on surpassing dense merging performance, which it appears the authors later downplayed in the rebuttal. While the reviewers acknowledge the methodology and findings of the paper (especially post-rebuttal) are interesting and experimental methodology overall sound, the wording of claims do matter when evaluating if the paper provides sufficient evidence for them. The work is clearly not a "paradigm shift" in the typical usage of that phrase within the scientific community, and unfortunately these over-claims do weaken what is otherwise I believe a concrete contribution that will be of interest to researchers within the community on low-rank model merging.